# Nucleoporin Nup155 is part of the p53 network in liver cancer

Kerstin Holzer[1,2], Alessandro Ori[3,4], Amy Cooke[5], Daniel Dauch[6,7], Elisabeth Drucker[1], Philip Riemenschneider[2], Amparo Andres-Pons[4,8], Amanda L. DiGuilio[9,10], Marie-Therese Mackmull[4], Jochen Baßler [11], Stephanie Roessler [1], Kai Breuhahn[1], Lars Zender[6,7], Joseph S. Glavy[9,12], Frank Dombrowski[2], Ed Hurt [11], Peter Schirmacher[1], Martin Beck [4,13] & Stephan Singer[1,2,4]

Cancer-relevant signalling pathways rely on bidirectional nucleocytoplasmic transport events through the nuclear pore complex (NPC). However, mechanisms by which individual NPC components (Nups) participate in the regulation of these pathways remain poorly understood. We discover by integrating large scale proteomics, polysome fractionation and a focused RNAi approach that Nup155 controls mRNA translation of p21 (*CDKN1A*), a key mediator of the p53 response. The underlying mechanism involves transcriptional regulation of the putative tRNA and rRNA methyltransferase FTSJ1 by Nup155. Furthermore, we observe that Nup155 and FTSJ1 are p53 repression targets and accordingly find a correlation between the p53 status, Nup155 and FTSJ1 expression in murine and human hepatocellular carcinoma. Our data suggest an unanticipated regulatory network linking translational control by and repression of a structural NPC component modulating the p53 pathway through its effectors.

[1] Institute of Pathology, University Hospital Heidelberg, 69120 Heidelberg, Germany. [2] Institute of Pathology, University Medicine Greifswald, 17475 Greifswald, Germany. [3] Leibniz-Institute on Aging, Fritz-Lipmann-Institute (FLI), 07745 Jena, Germany. [4] European Molecular Biology Laboratory, Structural and Computational Biology Unit, 69117 Heidelberg, Germany. [5] European Molecular Biology Laboratory, Directors' Research Unit, 69117 Heidelberg, Germany. [6] Department of Internal Medicine VIII, University Hospital Tuebingen, 72076 Tuebingen, Germany. [7] Translational Gastrointestinal Oncology Group, German Consortium for Translational Cancer Research (DKTK), German Cancer Research Center (DKFZ), 69120 Heidelberg, Germany. [8] Friedrich Miescher Institute for Biomedical Research, 4058 Basel, Switzerland. [9] Department of Chemistry, Chemical Biology and Biomedical Engineering, Stevens Institute of Technology, Hoboken, NJ 07030, USA. [10] Department of Biochemistry and Molecular Biology, The University of Chicago, Chicago, IL 60637, USA. [11] Heidelberg University Biochemistry Center, 69120 Heidelberg, Germany. [12] Department of Pharmaceutical Sciences, Ben and Maytee Fisch College of Pharmacy, University of Texas at Tyler, Tyler, TX 75799, USA. [13] Max Planck Institute of Biophysics, 60438 Frankfurt am Main, Germany. Correspondence and requests for materials should be addressed to S.S. (email: stephan.singer@uni-greifswald.de)

The p53 signalling pathway serves as a major barrier against the development and progression of cancer. As a transcription factor, p53 transactivates subsets of its myriad target genes in response to a variety of stresses that include DNA-damage, hypoxia, oncogene activation, and telomere shortening, among others[1,2]. A pivotal p53 target involved in mediating several cellular outcomes such as senescence and cell cycle arrest is the cyclin-dependent kinase inhibitor (CDKN1A), p21[1,3]. Thus, in this context p21 is considered to exhibit tumor suppressive functions[4]. However, p21 can also act in a protumorigenic fashion for instance by inhibition of caspases and therefore plays a context-dependent, dual role in cancer biology[4]. The complexity of p21's functions is probably exceeded by the complexity of mechanisms controlling its expression at various transcriptional, post-transcriptional, and post-translational levels[5]. At the level of p21 (CDKN1A) mRNA translation particularly those mechanisms involving the 5′UTR and/or the 3′UTR in the context of miRNAs or RNA-binding proteins (RBPs) such as CUG-binding protein1 (CUGBP1) and Calreticulin[6] or Musashi-1[7] have been studied.

The human NPC is a 110 MDa structure composed of ~30 different proteins, which form subcomplex units[8–10]. These units assemble in multiple copies to furnish an intricate machine that implements macromolecular nucleocytoplasmic transport. Nucleoplasmic, cytoplasmic and inner ring (IR) complexes constitute a structural scaffold that is complemented by flexible peripheral modules including phenylalanine rich filaments, a nuclear basket and transport channel Nups[8]. Accordingly, Nups are dichotomously recognized as structured scaffolding proteins or peripheral NPC components. Nup155 localizes to the IR region and along with Nups: 205, 188, 93, and 35 composes the mammalian IR complex[11,12]. The latest structural studies of NPC scaffold architecture reveal that certain copies of Nup155 interact directly with the nuclear envelope and additionally occur as rod-like connectors between the IR complex and the Y complex, a widely studied scaffold module[12,13].

Recent evidence confirms that peripheral NPC components have functions independent of transport. For example, Nup98 has been shown to influence chromosomal organization and regulation of gene expression, predominantly in *Drosophila melanogaster*[14,15]. In fact, we have previously demonstrated that Nup98 regulates multiple p53 target genes related to stress response and cell cycle control, namely p21 (CDKN1A) and 14–3–3 σ (SFN)[16]. In contrast, the ability of structural Nups (e.g. Nup155) to participate in selective gene regulation in a disease-relevant context is vastly unstudied. Surprisingly, based on its scaffold localization within the NPC[11,12], Zhang et al. demonstrated that a mutant version of Nup155 can cause a specific phenotype in mice and humans, which is linked with atrial fibrillation[17]. The authors could associate this phenotype to an altered HSP70 mRNA and protein transport[17]. Interestingly, a transcriptomic classification study of hepatocellular carcinoma (HCC)[18], one of the most prevalent malignancies[19], identified a HCC subgroup showing high expression of NUP155. This subgroup (G3) was further characterized by TP53 mutations and overexpression of cell cycle regulating genes[18]. However, the functional/mechanistic link between the p53 pathway and Nup155 remains unknown.

Here we show that Nup155 is required for full translation of p21 (CDKN1A) mRNA (independent of its 3′ and 5′UTR) involving the putative tRNA and rRNA methyltransferase FTSJ1. We also demonstrate that NUP155 and FTSJ1 are targets of p53-mediated repression giving rise to overexpression of both factors in murine and human HCC upon loss of p53's wild-type function.

## Results

**Full induction of p21 is Nup155-dependent.** To study the interplay between Nup155 and the p53 pathway in HCC we first tested if Nup155 is required for p53 target gene activation. To do so, we transfected HepG2 cells, harboring wild-type p53, with two different Nup155 siRNAs or a control siRNA (Allstars, AS) followed by a 24 h treatment with Nutlin-3a. Through disruption of the mdm2-p53 interaction, Nutlin-3a stabilizes p53 by protecting it from degradation leading to p53 target gene activation. Global protein abundance (~3500 proteins) was measured for the two Nup155 knockdown conditions normalized to siRNA controls (AS) by peptide-labeled mass spectrometry (Supplementary Data 1). Interestingly, the vast majority of proteins were not strikingly altered and only ~4% showed a significant change (q-value < 0.1; Supplementary Data 1, highlighted in light blue). This indicated a specific rather than general change in the proteome upon Nup155 knockdown. Among measured bona fide p53 targets (n = 48, Fig. 1a, red dots, Supplementary Table 1) we found p21 (Fig. 1a, green dot) to be less induced upon Nup155 knockdown, while other important p53 targets such as 14–3–3 σ and bax remained unchanged. To validate the impact of Nup155 depletion on p21 induction we prepared samples of the corresponding conditions including a third Nup155 siRNA for conventional immunoblotting. Figure 1b (upper and lower panel) illustrates that p21 protein was significantly decreased in the absence of Nup155 in Nutlin-3a treated HepG2 cells, while induced p53 levels were unchanged (for densitometric quantification see also Supplementary Fig. 1A). Notably, knockdown of other members of the Nup155-containing IR complex such as Nup35, Nup93, and Nup188 did not reduce p21 induction (Fig. 1c, Supplementary Fig. 1B–D). These data indicate that individual Nups of the same subcomplex can act distinctly in the p53 pathway. Importantly, we observed no striking impact on the abundance of other NPC components (purple dots) upon Nup155 depletion (Fig. 1d, Supplementary Table 2). Furthermore, Nup155 knockdown did not result in overt NPC clustering based on MAb414 immunofluorescence (Fig. 1e), which is an indicator for the absence of structural and/or assembly defects[20]. The dependency of p21 induction on Nup155 was additionally observed in Hep3B-4Bv cells that harbor a temperature-sensitive p53 mutant[21], which exhibits wild-type activity at incubation temperatures of 32 degrees. Mdm2 as another p53 target remained unchanged (Supplementary Fig. 1E, F). These data suggest that among IR components Nup155 is selectively required for full induction of p21.

**Nup155 is involved in p21 mRNA translation.** Next, we focused on dissecting the mechanism by which Nup155 regulates p21 due to its role as a key effector of several p53 responses. We utilized an H1299 derivative cell line expressing p21 (CDKN1A) from an artificial Tetracycline (Tet)-sensitive cDNA construct (H24-p21)[22] to determine whether the observed effect is dependent on or independent of the endogenous CDKN1A promoter. We could show that exogenously expressed p21, following Tet removal (Tet-off), was significantly reduced upon Nup155 knockdown (Fig. 2a, Supplementary Fig. 2A). Furthermore, the corresponding p21 (CDKN1A) mRNA levels remained unchanged following Nup155 depletion, as measured by qRT-PCR (Fig. 2b). We thus concluded that the observed phenotype is both independent of the endogenous CDKN1A promoter and of p21 (CDKN1A) mRNA transcription. Nup155 was previously reported to be involved in HSP70 mRNA export and HSP70 protein import[17]. We therefore investigated if decreased p21 protein accumulation upon Nup155 knockdown can be attributed to a p21 (CDKN1A) mRNA export block in HepG2 cells. Consistent with previously

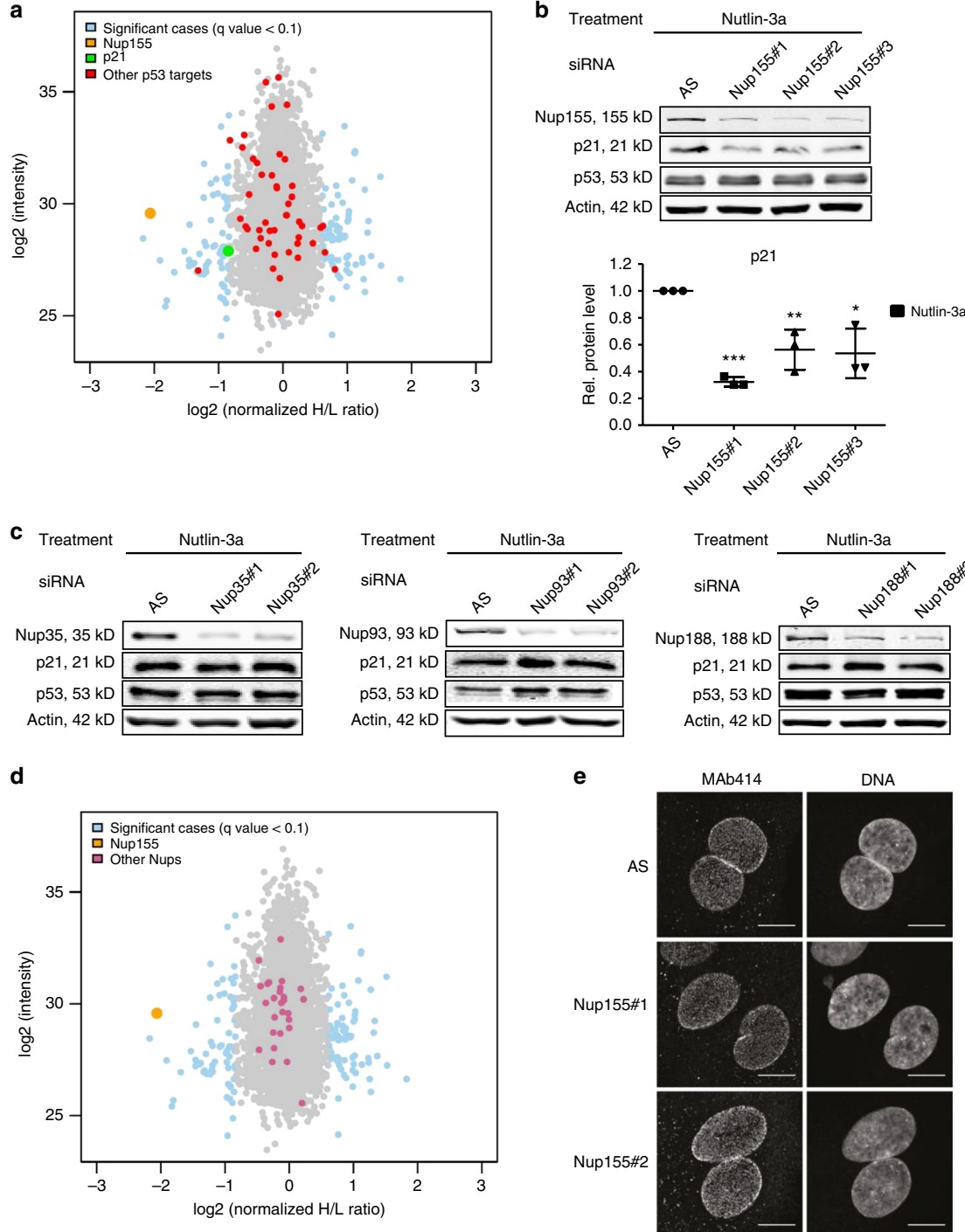

**Fig. 1** Full induction of p21 is Nup155-dependent. **a** Protein fold-changes (blue and grey dots) including p53 targets (red dots) upon Nup155 (orange dot) knockdown compared to the control siRNA (AS) condition are shown as log2 ratios. p21 is highlighted in green as a key effector of the p53 response. Overall 3523 proteins could be quantified by quantitative mass spectrometry in HepG2 cells treated with Nutlin-3a for 24 h. **b** HepG2 cells were treated either with control siRNA (AS) or three different Nup155 siRNAs (Nup155#1, Nup155#2, and Nup155#3) for 72 h and p53 was induced by adding Nutlin-3a for 24 h. Cell extracts were analysed by immunoblotting with indicated antibodies (upper panel). Densitometric quantification of immunoblots derived from three independent experiments is shown in the lower panel and normalised to the control siRNA (AS) condition. **c** HepG2 cells were treated either with control siRNA (AS) or two different Nup35, Nup93, and Nup188 siRNAs (Nup35#1, Nup35#2, Nup93#1, Nup93#2, Nup188#1, and Nup188#2) for 72 h. Cells were harvested upon 24 h of Nutlin-3a treatment and extracts were analysed by immunoblotting with the indicated antibodies. For corresponding densitometric analyses see Supplementary Figure 1B-D. **d** Protein fold-changes of NPC components (purple dots) upon Nup155 knockdown (orange dot) corresponding to the conditions described in **a**. **e** Representative confocal microscopy pictures of MAb414 immunofluorescence staining of HepG2 cells treated as indicated. DAPI was used for DNA labeling. Scale bar = 10 $\mu$m. *$p < 0.05$, **$p < 0.01$, ***$p < 0.001$ (Student's $t$-test). Data are presented as mean ± stdv. Source data are provided as a Source Data file

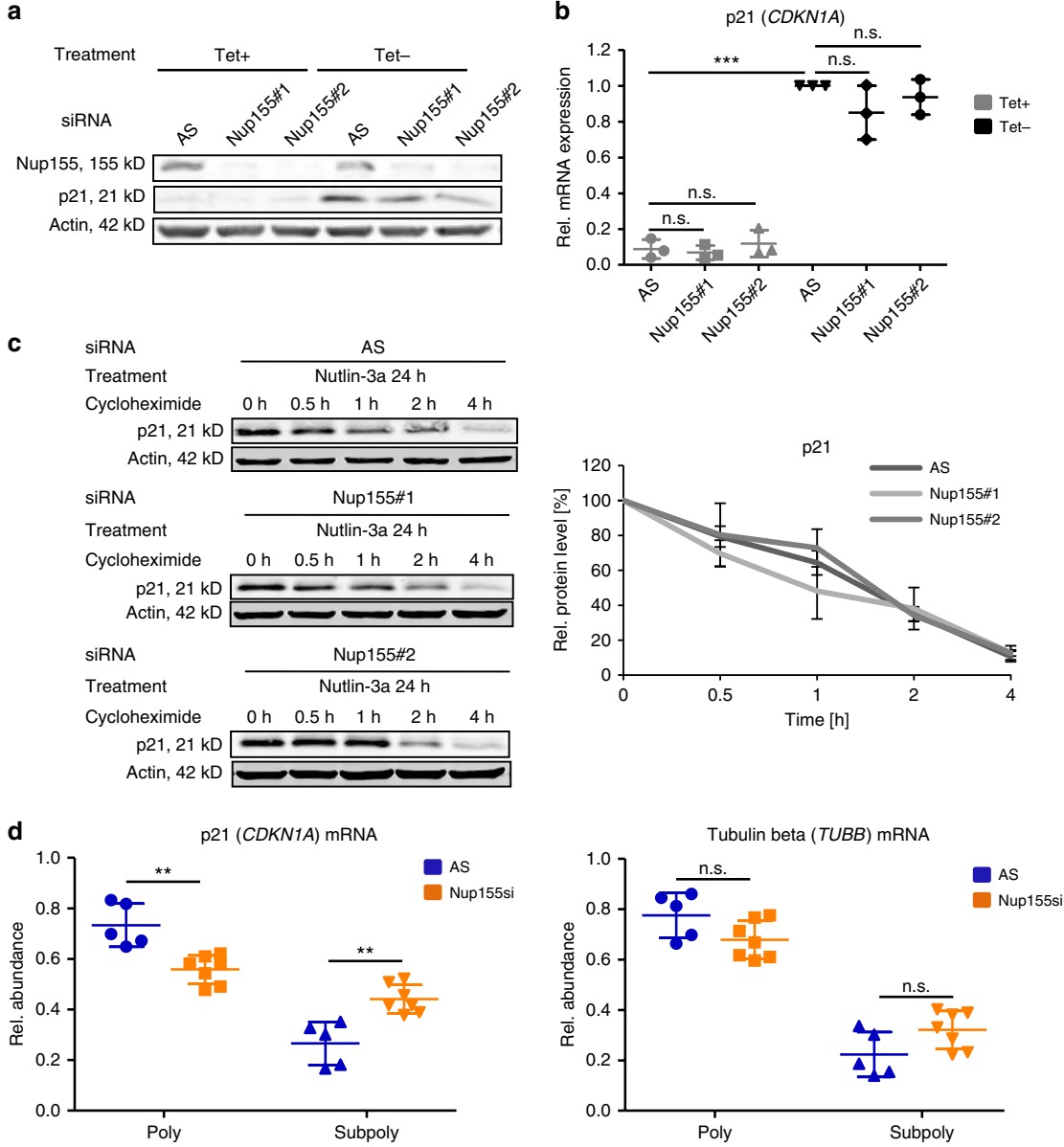

**Fig. 2** Nup155 is involved in p21 mRNA translation. **a** H24-p21 cells containing a tetracycline (Tet-off)-regulatable p21 expression construct were either treated with control siRNA (AS) or two different Nup155 siRNAs (Nup155#1 and Nup155#2) for 72 h. p21 was induced by tetracycline (Tet) removal 96 h before harvesting. Cell extracts were analysed by immunoblotting with the indicated antibodies. For corresponding densitometric analyses see Supplementary Figure 2A. **b** Relative p21 (*CDKN1A*) mRNA levels of the corresponding conditions described in **a** were measured by qRT-PCR. Data are derived from three independent experiments and normalised to the Tet-off control siRNA (AS) condition. **c** HepG2 cells were treated either with control siRNA (AS) or two different Nup155 siRNAs (Nup155#1 and Nup155#2) for 72 h. p21 was induced by adding Nutlin-3a 24 h before blocking translational elongation with cycloheximide. Cells were harvested at indicated time points and analysed by immunoblotting with the indicated antibodies (left panels). Densitometric quantification analyses of the above-mentioned p21 half-life experiments normalised to the control siRNA condition (right panel). **d** HepG2 cells were treated either with control siRNA (AS) or two different Nup155 siRNAs for 72 h. Translational elongation was blocked with cycloheximide and polyribosome analyses were performed. p21 (left panel) and tubulin beta mRNA (right panel) of the polysomal (poly) and subpolysomal (subpoly) fractions were measured by qRT-PCR with the expression levels normalised to Renilla spike-in RNA. *p*-values result from the comparison of the control (AS) ($n = 5$) and combined Nup155 siRNAs ($n = 7$), see also Methods section. **$p < 0.01$, ***$p < 0.001$ (Student's *t*-test); Data are presented as mean ± stdv. Source data are provided as a Source Data file

published data, subcellular fractionation followed by qRT-PCR revealed that there was a significant decrease in the cytoplasmic/nuclear ratio of *HSP70* mRNA upon Nup155 knockdown (Supplementary Fig. 2b, right panel). However, no significant change of the cytoplasmic/nuclear ratio of p21 (*CDKN1A*) mRNA was detectable (Supplementary Fig. 2B, left panel) excluding a substantial export defect of p21 (*CDKN1A*) mRNA. We also did not observe an obvious change in the subcellular distribution of p21

protein between the aforementioned conditions as analysed by immunocytochemistry (Supplementary Fig. 2C). To determine if Nup155 affected p21 protein turnover we performed cycloheximide chase experiments in HepG2 cells, in which mRNA translation is blocked and thereby allows protein degradation over time to be monitored. Immunoblotting and corresponding densitometric quantification (Fig. 2c) confirmed that p21 protein half-life was not affected by Nup155-siRNA treatment. The same

result was obtained using the above-mentioned p21-inducible cells (Supplementary Fig. 2D).

We then hypothesized that Nup155 might be directly or indirectly involved in p21 (*CDKN1A*) mRNA translation. Examination of differences in ribosomal occupancy of p21 (*CDKN1A*) mRNA in the presence and absence of Nup155 was conducted by polysome fractionation experiments and subsequent qRT-PCR in HepG2 cells. Following separation by sucrose gradient, polysomal and subpolysomal mRNA fractions were pooled prior to RNA extraction and analysed by qRT-PCR to assess the levels of translationally active versus inactive transcripts, respectively. The pooled fractions are illustrated by absorbance profiles for control (AS) and the two Nup155-siRNA treatment conditions (Supplementary Fig. 2E). Notably, the amount of p21 (*CDKN1A*) mRNA in the polysomal fraction or translationally active fraction significantly decreased ($p < 0.01$, (Student's $t$-test)) in the combined Nup155 knockdown conditions as compared to the control siRNA (AS) condition (Fig. 2d, left panel). Accordingly, the amount of p21 (*CDKN1A*) mRNA in the subpolysomal fraction significantly increased ($p < 0.01$, (Student's $t$-test)) following Nup155 depletion, as shown in Fig. 2d (left panel). Conversely, tubulin beta (*TUBB*) mRNA showed no significant difference in the respective fractions following Nup155 knockdown (Fig. 2d, right panel). Collectively, these data suggest that Nup155 is required for p21 mRNA translation.

**Nup155 regulates p21 independent of miRNAs and its 3′/5′-UTR.** A variety of miRNAs have been documented to control p21 (*CDKN1A*) mRNA stability and translation[5]. We therefore tested if the observed phenotype can be rescued by blocking miRNA biogenesis. We thus subjected HepG2 cells to DROSHA knockdown (DROSHA#1 and DROSHA#2) in the presence or absence of Nup155 siRNAs under Nutlin-3a treatment (Supplementary Fig. 3A). DROSHA depletion resulted in an increase of p21 protein, presumably as a result of increased p21 (*CDKN1A*) mRNA levels and p21 (*CDKN1A*) mRNA translation. However, co-depletion of Nup155 and DROSHA decreased these elevated p21 protein levels (Supplementary Fig. 3A), indicating that p21 is still sensitive to Nup155 knockdown even under impaired miRNA biogenesis conditions. We conclude that regulation of p21 (*CDKN1A*) mRNA translation by Nup155 occurs in a miRNA-independent manner.

Still, a number of RNA-binding proteins (RBPs) have been described to influence mRNA translation. These include CUGBP1, HuR, Musashi, and hRNPs, among others and typically bind to regulatory regions within the 3′UTR or 5′UTR of p21 (*CDKN1A*) mRNA[5]. To determine whether these regions of mRNA are required for the observed phenotype, we transfected H1299 cells with full-length p21 (p21FL) or a construct lacking the 3′UTR (p21Δ3′UTR) or the 3′UTR and the 5′UTR of p21 (p21Δ3′Δ5′UTR) in the presence or absence of Nup155. Figure 3a–c illustrate that exogenously expressed p21 protein was sensitive to Nup155 knockdown irrespective of the 3′UTR or 5′UTR sequences. A GFP-expressing construct served as a negative control (Fig. 3d). Taken together, the requirement of Nup155 for p21 (*CDKN1A*) mRNA translation could not be abrogated by removing the 5′UTR and/or the 3′UTR of p21 (*CDKN1A*) mRNA. Identification of potential mediators for Nup155-dependent p21 (*CDKN1A*) mRNA translation was achieved by further analyses of the proteomic dataset acquired upon Nup155 knockdown (Fig. 1a and Supplementary Data 1). Candidates implicated in mRNA translation such as Zinc Finger Protein 385 A (Z385A), tRNA-splicing endonuclease subunit Sen15 (SEN15), 60 S ribosomal protein L7-like 1 (RL7L1),

Nucleolar GTP Binding Protein 1 (NOG1) and the putative tRNA methyltransferase FTSJ1 (FTSJ1), were selected from the subgroup of significantly downregulated proteins after Nup155 depletion and subsequently included in a focused RNAi approach (Fig. 3e). Using two specific siRNAs for each candidate, we tested if the direct knockdown of any of these factors could recapitulate the phenotype observed upon Nup155 depletion. Out of this group only the putative tRNA methyltransferase, FTSJ1 (human homologue of yeast TRM7), phenocopied Nup155 depletion by exhibiting reduced p21 protein accumulation in Nutlin-3a treated HepG2 cells without affecting p53 induction in both knockdown conditions (Fig. 3e). Immunoblotting confirmed that the FTSJ1 protein is reduced on protein level upon Nup155 knockdown (Fig. 3f, Supplementary Fig. 3B).

**FTSJ1 is involved in p21 (CDKN1A) mRNA translation.** We further validated FTSJ1 as relevant candidate by showing that reduced p21 protein accumulation (Fig. 4a upper and middle panel, Supplementary Fig. 4A) is not reflected at p21 (*CDKN1A*) transcript level (Fig. 4a, lower panel). Moreover, we could demonstrate by polysome fractionation that FTSJ1 depletion leads to significantly decreased polysomal and increased subpolysomal p21 (*CDKN1A*) mRNA ($p < 0.05$, respectively, (Student's $t$-test)), while tubulin beta (*TUBB*) mRNA levels remain unchanged (Fig. 4b, upper and lower panel, respectively). The corresponding absorbance profiles for control and FTSJ1 siRNAs are shown in Supplementary Fig. 4B.

The physiologically relevant tRNA target of the TRM7 containing complex in yeast is the tRNA[Phe][23]. We examined if alternative codon usage for phenylalanine in p21 would abolish its dependency on Nup155. However, exchanging TTC codons to TTT in the p21 (*CDKN1A*) CDS (Supplementary Fig. 4C, left panel)) did not reverse the reduction of p21 protein in the absence of Nup155 (Supplementary Fig. 4C, middle and right panel, Supplementary Fig. 4D, E). We thus conclude that Nup155 regulates p21 (*CDKN1A*) mRNA translation through FTSJ1, independent of the TTC codon usage and (besides other conceivable mechanisms including protein-protein interactions) possibly by its putative role in ribosomal RNA methylation.

**Nup155-dependent regulation of FTSJ1 is linked to HDAC4.** To determine if Nup155 regulates FTSJ1 at the level of transcription we tested if the decrease of FTSJ1 protein upon Nup155 knockdown is paralleled by reduced *FTSJ1* mature and nascent (unspliced) mRNA. To do so, we performed qRT-PCR analyses using exon–exon spanning primer pairs to probe for mature and intron–exon spanning primer pairs to detect nascent *FTSJ1* mRNA. As shown in Fig. 5a, we observed that both, mature and nascent *FTSJ1* mRNA levels were significantly lower in the Nup155-siRNA conditions suggesting a regulatory mechanism at the transcriptional level. It was previously reported that Nup155 is an HDAC4-interacting protein and that this interaction was required for HDAC4-induced gene expression patterns[24]. We therefore hypothesized that Nup155-dependent regulation of *FTSJ1* involves HDAC4. While the aforementioned study used cardiomyocytes for the respective analyses, we first tested if the interaction of both proteins can be recapitulated in HepG2 cells. Indeed, we were able to co-immunoprecipitate HDAC4 and Nup155 confirming an interaction of both proteins also in liver-derived cancer cells (Supplementary Fig. 5A). Furthermore, HDAC4 depletion by using two different siRNAs was followed by a significant decrease of nascent *FTSJ1* mRNA as measured by qRT-PCR (Fig. 5b). Conversely, co-overexpression of HDAC4 and Nup155 resulted in a significant increase of *FTSJ1* mRNA (Fig. 5c, Supplementary Fig. 5B). We conclude that

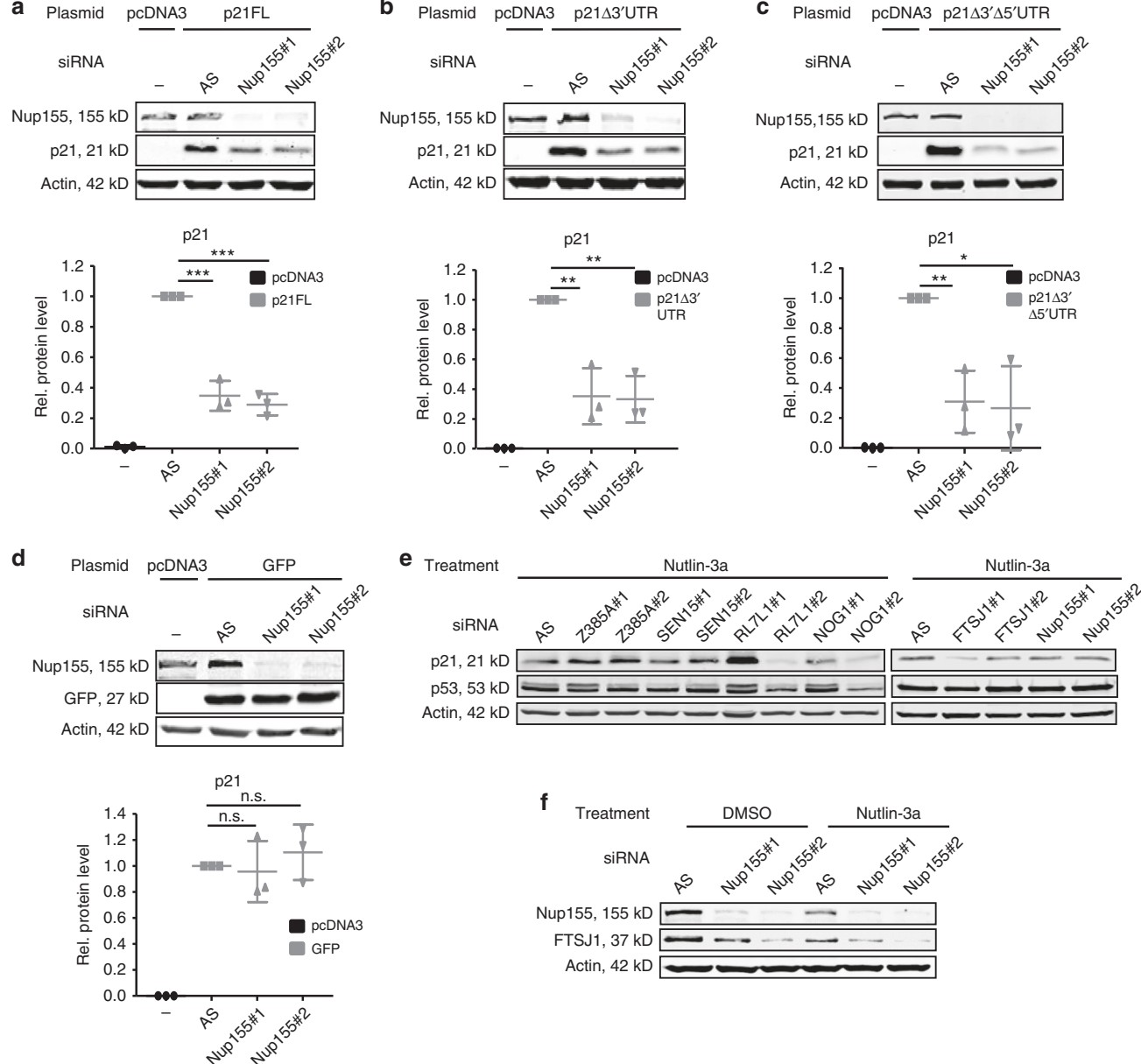

**Fig. 3** Nup155-dependent p21 mRNA translation is independent of the 3′/5′UTR and potentially involves FTSJ1. **a–d** H1299 cells were treated either with control (AS) or two different Nup155 siRNAs (Nup155#1 and Nup155#2) for 72 h and either co-transfected with a control vector (pcDNA3) or different p21 expression constructs with p21 full-length (p21FL) (**a**) or constructs that lack either the 3′UTR (p21Δ3′UTR) (**b**) or the 5′UTR and the 3′UTR (p21Δ3′Δ5′UTR) (**c**). A GFP-expressing construct served as a negative control (**d**). Cell extracts were analysed by immunoblotting with the indicated antibodies (upper panels) and corresponding densitometric analyses derived from three independent experiments are shown in the lower panels and are normalised to the control siRNA condition. **e** Candidates with a suggested role in mRNA translation and downregulated upon Nup155 knockdown as revealed by the proteomic approach (see Fig. 1A) were included in a focused RNAi approach. Reduced p21 protein accumulation and unaltered p53 levels as assayed by immunoblotting were considered as phenocopy of Nup155 depletion. Corresponding immunoblots show p21 and p53 levels (under Nutlin-3a treatment) either from the control siRNA (AS) condition or the candidate knockdown conditions (using two different siRNAs #1 and #2). Actin served as loading control. **f** HepG2 cells were treated either with control siRNA (AS) or two Nup155 siRNAs (Nup155#1 and Nup155#2) for 72 h and p53 was induced by adding Nutlin-3a for 24 h. Cell extracts were analysed by immunoblotting with indicated antibodies. For densitometric analyses see Supplementary Figure 3B. *$p < 0.05$, **$p < 0.01$, ***$p < 0.001$ (Student's $t$-test); Data are presented as mean ± stdv. Source data are provided as a Source Data file

transcriptional regulation of *FTSJ1* by Nup155 involves its interaction partner HDAC4.

**Nup155 and FTSJ1 are targets of p53-mediated repression**. As another important aspect of the crosstalk between Nup155, FTSJ1 and the p53 pathway, we noticed that Nup155 and FTSJ1 proteins are reduced after p53 induction (Fig. 3f, lane 1 vs. lane 4,

respectively). To substantiate this observation, we treated another p53 wild-type containing cancer cell line (Sk-Hep1) with Nutlin-3a for 24 and 48 h. Figure 6a demonstrates also in Sk-Hep1 cells a gradual decrease of Nup155 and FTSJ1 protein (upper and middle panel) upon p53 accumulation. The decrease of Nup155 and FTSJ1 proteins were paralleled by reduced nascent *NUP155* and *FTSJ1* mRNA suggesting transcriptional repression by p53

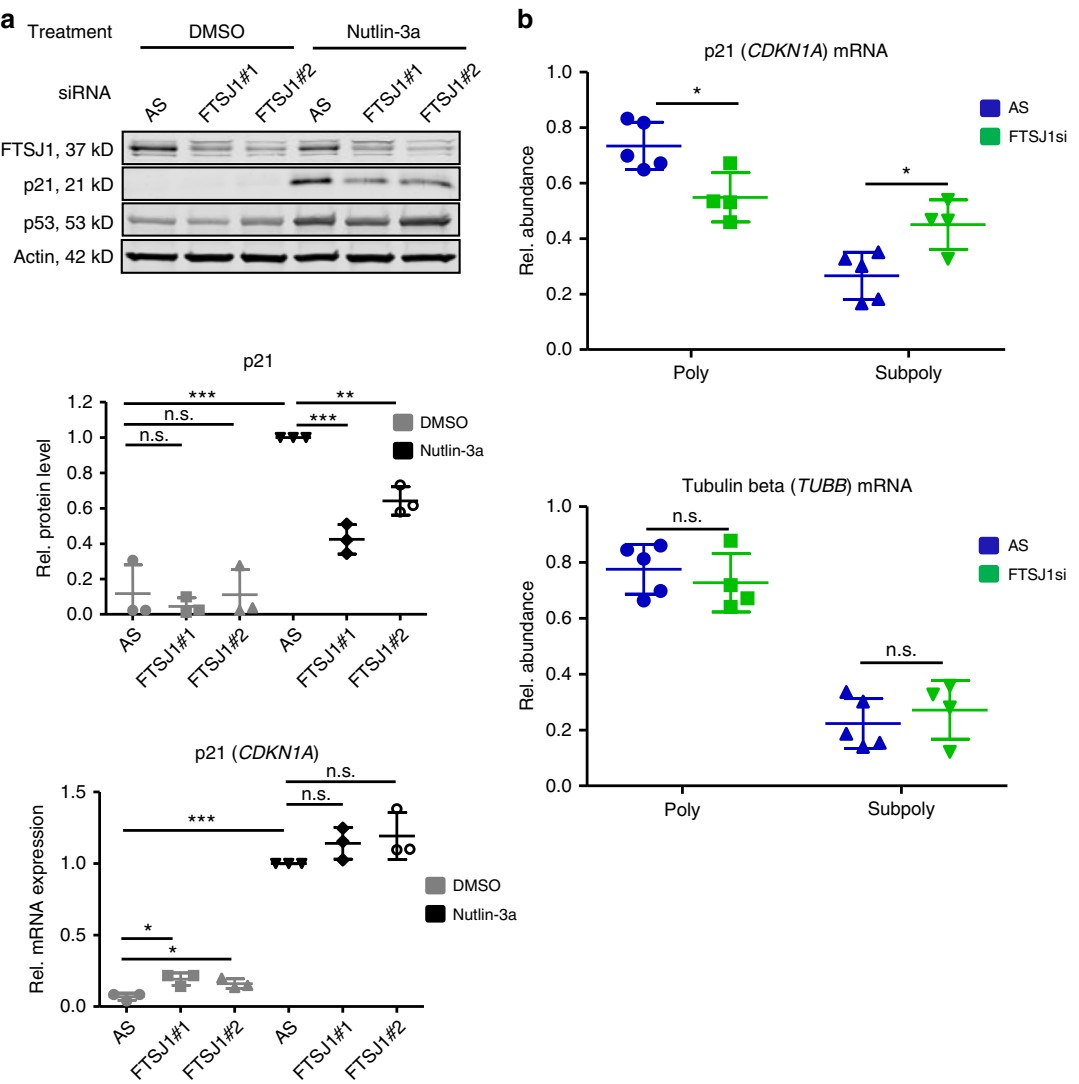

**Fig. 4** FTSJ1 knockdown phenocopies Nup155 depletion. **a** Upper panel: HepG2 cells were treated either with control siRNA (AS) or two FTSJ1 siRNAs (FTSJ1#1 and FTSJ1#2) for 72 h and p53 was induced by adding Nutlin-3a for 24 h. Cell extracts were analysed by immunoblotting with indicated antibodies. Middle panel: densitometric quantification analyses of p21 immunoblots derived from three independent experiments normalised to the Nutlin-3a control siRNA condition. For densitometric analyses of FTSJ1 and p53 see Supplementary Figure 4 A. Lower panel: relative p21 (*CDKN1A*) mRNA levels of the corresponding conditions described above were measured by qRT-PCR. Data are derived from three independent experiments and normalised to the Nutlin-3a control siRNA condition. **b** HepG2 cells were treated either with control siRNA (AS) or two different FTSJ1 siRNAs (FTSJ1#1 and FTSJ1#2) for 72 h. Polyribosome analyses were performed as described in Fig. 2d. p21 (*CDKN1A*) mRNA (left panel) and tubulin beta (*TUBB*) mRNA (right panel) of the polysomal (poly) and subpolysomal (subpoly) fractions were measured by qRT-PCR with the expression levels normalized to Renilla spike-in RNA. *p*-values result from the comparison of the control (AS) (*n* = 5) and combined FTSJ1 siRNAs (*n* = 4). The control siRNA (AS) condition is identical to that shown in Fig. 2d (see also Methods section). *$p < 0.05$, **$p < 0.01$, ***$p < 0.001$ (Student's *t*-test); Data are presented as mean ± stdv. Source data are provided as a Source Data file

(Fig. 6a lower panel). Importantly, lowered *NUP155* and *FTSJ1* mRNA and protein upon Nutlin-3a treatment could be rescued by siRNA-mediated depletion of p53 (Fig. 6b, Supplementary Fig. 6A). Transcriptional repression by p53 frequently occurs in a p21-dependent manner[25], thus we evaluated if p21 induction is required for Nup155 and FTSJ1 downregulation in this setting. Indeed, knockdown of p21 by RNA*i* largely rescued reduced *NUP155* and *FTSJ1* mRNA upon p53 induction (Fig. 6c). These data suggest that *NUP155* and *FTSJ1* are repressed by wild-type p53 in a p21-dependent manner.

Since targets that are repressed by wild-type p53 can be induced by gain-of-function p53 mutants[26], we tested if Nup155 and FTSJ1 are sensitive to knockdown of mutant p53 in HCC cell

lines (HuH7, HLE and HLF). However, Nup155 and FTSJ1 protein remained unaltered upon depletion of the corresponding endogenously expressed p53 mutants p53[Y220C] and p53[G244A] (Supplementary Fig. 6B-D)). We conclude that Nup155 and FTSJ1 do not represent gain-of-function targets of the aforementioned p53 mutants.

**Nup155 and FTSJ1 expression correlate with p53 status in HCC.** Based on these data, we hypothesized that HCCs harbouring wild-type p53 express lower levels of both Nup155 and FTSJ1. To test this hypothesis in vivo, we analysed murine HCCs generated by transposon-based gene transfer of Myc and N-ras[G12V] into mice with homozygous deletion of Trp53 (Trp53

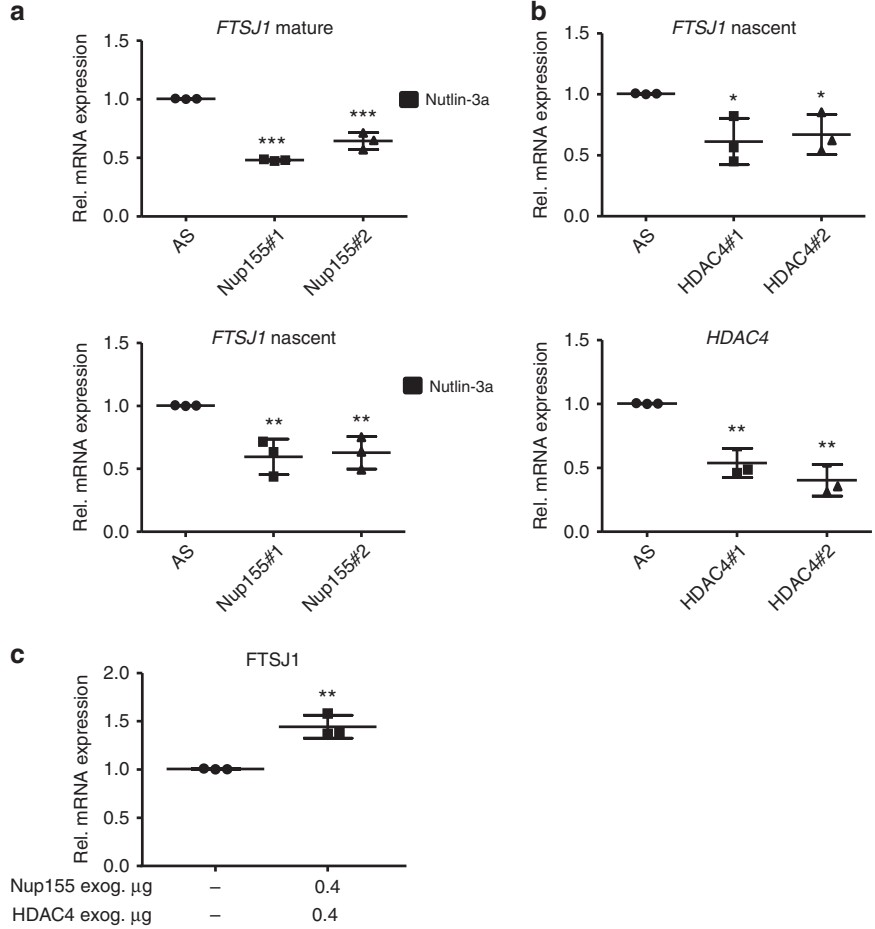

**Fig. 5** HDAC4 is involved in FTSJ1 transcriptional regulation. **a** Relative mature (upper panel) and nascent (lower panel) mRNA levels of *FTSJ1* in HepG2 cells upon Nup155 knockdown and Nutlin-3a treatment for 24 h as measured by qRT-PCR. Data are derived from three independent experiments and normalised to the control siRNA condition. **b** HepG2 cells were treated either with control siRNA (AS) or two different HDAC4 siRNAs (HDAC4#1 and HDAC4#2) for 72 h and p53 was induced by adding Nutlin-3a for 24 h. Relative nascent mRNA levels of *FTSJ1* (upper panel) and *HDAC4* (lower panel) as measured by qRT-PCR. Data are derived from three independent experiments and normalised to the control siRNA (AS) condition. **c** HepG2 cells were co-transfected with *HDAC4*-Flag and *Nup155*-HA constructs for 24 h. Empty vector control (−). Relative *FTSJ1* mRNA level as measured by qRT-PCR. Data are derived from three independent experiments and normalised to the control vector condition. *$p < 0.05$, **$p < 0.01$, ***$p < 0.001$ (Student's *t*-test); Data are presented as mean ± stdv. Source data are provided as a Source Data file

−/−), the murine homolog of human p53, or into wild-type mice (Trp53+/+)[27]. Figure 7a demonstrates that Nup155 and FTSJ1 were more strongly expressed in Trp53−/− HCCs compared to Trp53+/+HCCs. We also evaluated *NUP155* and *FTSJ1* gene expression in a large cohort of HCC patients ($n = 247$). This analysis revealed that *NUP155* and *FTSJ1* were both over-expressed ($p < 0.001$, (Student's *t*-test)) in HCCs compared to the non-tumorous liver tissue (Fig. 7b, left and right panel) and positively correlated to each other ($r = 0.316$ (Pearson correlation coefficient); $p < 0.001$ (Pearson p-value)) (Fig. 7c). Higher than median expression of Nup155 was associated with poorer survival of HCC patients (Supplementary Fig. 7A, left panel), which was not the case for Nup188, another component of the Nup155-containing NPC subcomplex (Supplementary Fig. 7A, right panel). Finally, by comparing HCCs containing wild-type p53 with those harboring mutant p53 we observed significantly lower *NUP155* and *FTSJ1* transcript levels in the former as opposed to the latter (Supplementary Fig. 7B, left and middle panel). Notably, there was no significant expression difference for Nup188 (Supplementary Fig. 7B, right panel). Taken together, these murine and human HCC data are further consistent with *NUP155* and *FTSJ1* being targets of p53-mediated repression in vivo.

Based on our findings we propose a feedback loop model as illustrated in Fig. 7d. Upon p53 activation full p21 (*CDKN1A*) mRNA translation requires FTSJ1 being transcriptionally regulated via HDAC4 interacting with Nup155. At the same time Nup155 is repressed by p53 in a p21-dependent manner.

## Discussion

Increasing evidence indicates that members of the nuclear transport machinery can regulate cancer-relevant pathways at different levels such as import of transcription factors, chromatin interaction, and export/stability of target gene mRNAs[28]. In addition, Nup358/RanBP2 as a peripheral NPC component has also been linked to the level of mRNA translation[29,30]. A link between a scaffold Nup and transcriptional regulation of a gene involved in mRNA translation, as shown here for Nup155 has, to the best of our knowledge, not been described before in human cancer. Albeit, one of the yeast homologues of human Nup155, namely Nup170p, was shown to be part of the RSC chromatin remodeling complex and involved in transcriptional regulation of subtelomeric genes and genes encoding components of the translational machinery[31]. Here, we demonstrate that transcriptional regulation of *FTSJ1* by Nup155 involves its interaction partner HDAC4. This interaction has been shown

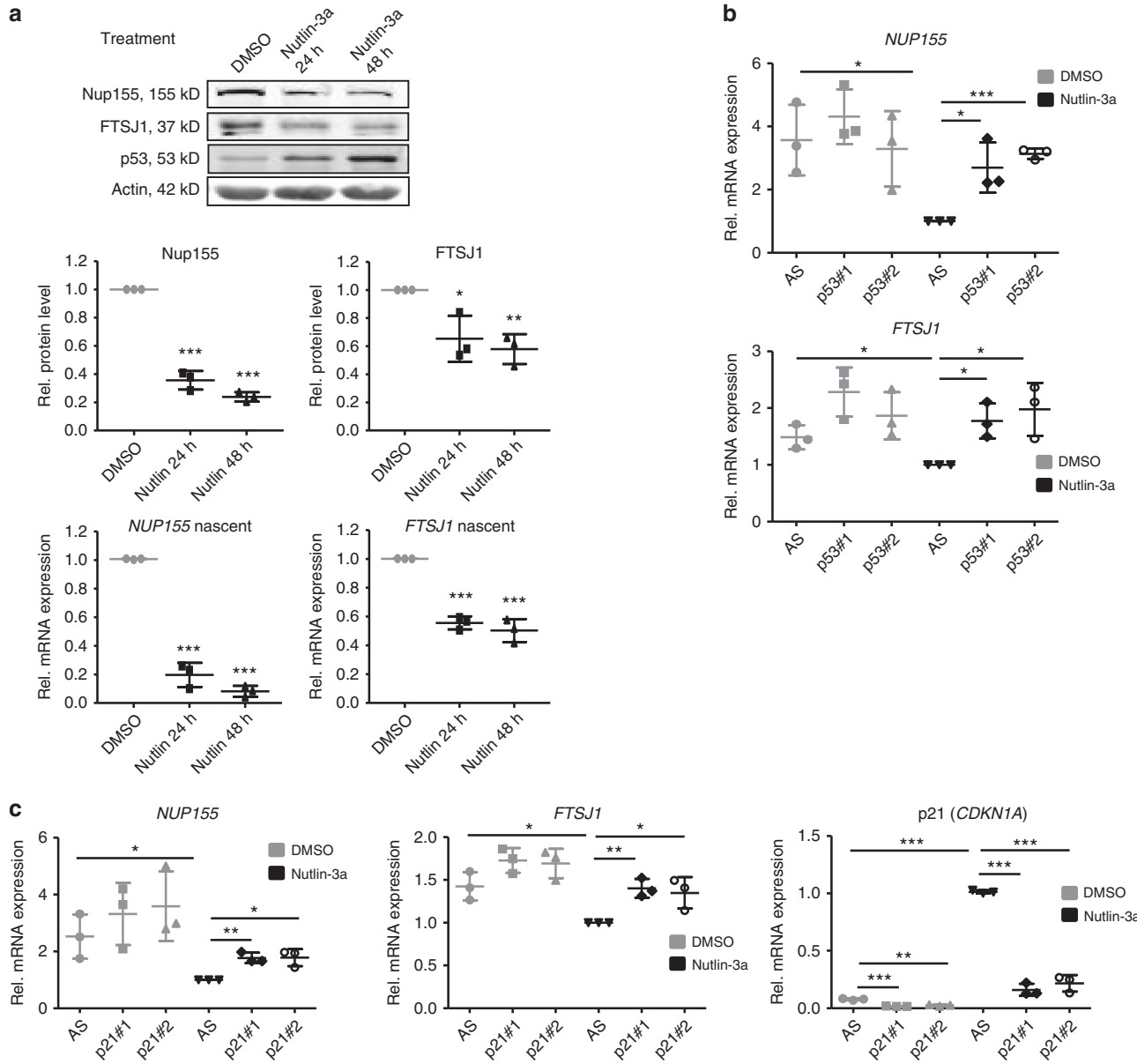

**Fig. 6** Nup155 and FTSJ1 are targets of p53-mediated repression. **a** Sk-Hep1 cells were treated either with DMSO or Nutlin-3a for 24 h and 48 h. Cell extracts were analysed by immunoblotting with indicated antibodies (upper panel). Nup155 and FTSJ1 densitometric analyses of immunoblots derived from three independent experiments (middle panels) normalised to the DMSO condition. Relative nascent mRNA levels of *NUP155* and *FTSJ1* as measured by qRT-PCR (lower panels). Data are derived from three independent experiments and normalised to the DMSO condition. **b** Sk-Hep1 cells were treated either with control siRNA (AS) or two different p53 siRNAs (p53#1 and p53#2) for 72 h. Cells were harvested upon 48 h of Nutlin-3a treatment and extracts were analysed by qRT-PCR (*NUP155* upper panel; *FTSJ1* lower panel). Data are derived from three independent experiments and normalised to the Nutlin-3a control siRNA condition. **c** Sk-Hep1 cells were treated either with control siRNA (AS) or two different p21 siRNAs (p21#1 and p21#2) for 72 h. Cells were harvested upon 24 h of Nutlin-3a treatment and extracts were analysed by qRT-PCR (*NUP155* left panel; *FTSJ1* middle panel; p21 (*CDKN1A*) lower panel). Data are derived from three independent experiments and normalised to the Nutlin-3a control siRNA condition. *$p < 0.05$, **$p < 0.01$, ***$p < 0.001$ (Student's t-test); Data are presented as mean ± stdv. Source data are provided as a Source Data file

before to be relevant for both HDAC4-dependent repression and activation of target genes at the NPC in the context of cardio-myocyte hypertrophy[24]. Interestingly, we did observe an increase in FTSJ1 expression when Nup155 and HDAC4 are simultaneously overexpressed. This could indicate that HDAC4 and Nup155 binding is saturated at steady state levels and thus overexpression of either binding partner is not sufficient to elevate *FTSJ1* expression. In contrast, Nup155 or HDAC4 knockdown alone results, respectively, in less Nup155-HDAC4 interaction and therefore less *FTSJ1* expression.

As one of the most important p53 targets p21 (*CDKN1A*) is tightly regulated at virtually all levels[5] including transcription initiation[3], elongation[32], mRNA degradation by miRNAs[33,34] and mRNA stabilization by RNA-binding proteins (RBP) interacting with the 3′UTR like HuR[35] or RNPC1[36]. In addition, post-translational modifications (e.g. phosphorylation by Akt1/PKB, PKA, PKC, and Pim-1[5] affecting protein localization and stability combined with ubiquitin-dependent[37,38] and -independent proteasomal degradation[39,40]) represent further well defined levels of regulation. Interestingly, regulatory mechanisms of p21

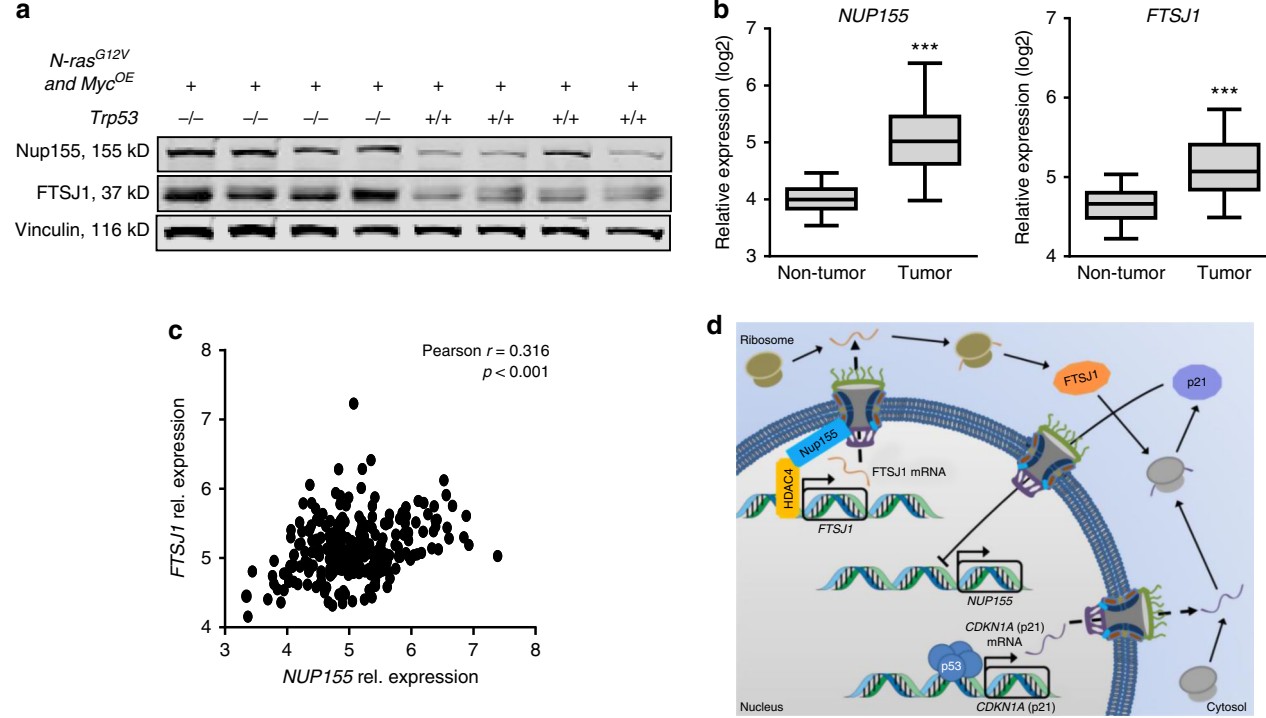

**Fig. 7** p53 status, Nup155 and FTSJ1 expression are correlated in murine and human HCC. **a** Murine *N-ras^G12V* *and Myc*-driven HCC which developed in a *Tpr53*-deleted (*Trp53^−/−*, n = 4) or -wild-type (*Trp53^+/+*, n = 4) background were analysed by immunoblotting with the indicated antibodies. **b** Relative transcript levels of *NUP155* (left panel) and *FTSJ1* (right panel), in HCC tissue (n = 247) compared to non-tumorous liver tissue (n = 239). The center line of the box plots denotes the median, the bounds of the box indicate the 25th to 75th percentiles and the whiskers represent the 5th to 95th percentiles, respectively. ***p < 0.001 (Student's t-test). **c** Correlation of *NUP155* and *FTSJ1* in HCC tissue (n = 247). r Pearson correlation coefficient, p Pearson p-value. **d** Suggested feedback loop model linking p53/p21 with Nup155 and FTSJ1. Source data are provided as a Source Data file

(*CDKN1A*) mRNA translation besides those involving the 5′UTR and/or the 3′UTR remain largely unstudied. Our co-transfection experiments with truncated p21 expressing constructs indicate, indeed, Nup155-mediated regulation of p21 to be independent of the 3′ and 5′UTR. Together with the finding that knockdown of the putative tRNA and rRNA methyltransferase FTSJ1 pheno-copies Nup155 depletion in terms of p21 (*CDKN1A*) mRNA translation our data suggest (among other mechanisms) rRNA methylation or codon usage to be potentially relevant. However, the latter appears less likely than the former, based on our codon exchange experiments for phenylalanine in the p21 CDS. In contrast, methylation of rRNA by FTSJ1 could be a viable sce-nario. Mainly provided by the rRNA 2′-O-methyltransferase fibrillarin (FBL) as part of a ribonucleoprotein complex (RNP), rRNA methylation plays an important role in translational con-trol[41]. Although further mechanistic analyses are required to link the putative rRNA methylation activity of FTSJ1 to p21 and the p53 pathway, the observation that (similar to FTSJ1) FBL also represents a p53 repression target[42] may already point in this direction. Nevertheless, it is also conceivable that the link between FTSJ1 and p21 (*CDKN1A*) mRNA translation is rather indirect via yet to be defined additional targets and/or interactors.

The data raised in murine HCCs indicate that genetic inacti-vation (=deletion) of Trp53 gives rise to high *NUP155* and *FTSJ1* expression in the respective tumors. In accordance, we found that *NUP155* and *FTSJ1* are transcriptionally repressed by wild-type p53. Hence, when comparing human tumor and non-tumorous liver samples overexpression of *NUP155* and *FTSJ1* in HCC is most likely due to a loss of p53's wild-type function followed by a de-repression in a considerable fraction of tumors, since inacti-vation of p53 (either functionally or genetically) frequently occurs

in human HCC[43]. The rather mild differences of NUP155 and FTSJ1 expression in *TP53* wild-type (wt) vs. *TP53* mutated (mut) HCCs could be explained by the fact that >90% of HCC patients in this cohort showed positivity for Hepatitis B Virus (HBV). The HBV-related protein HBx inhibits the wild-type function of p53[44–46] and therefore in these tumors p53 can be considered largely functionally inactive although being genetically wild-type.

Our data suggest a negative feedback loop between p21 and Nup155 apparently balancing p21 levels within the p53 response. It seems surprising that p21 induction is coupled to Nup155, which based on its expression profile and correlation with poorer patient outcome could be considered rather pro-tumorigenic. However, similar findings have been documented for another nuclear transport factor termed exportin-2 (XPO2/CAS). This transport factor was shown to bind to p53 target gene promoters and to be required for the full induction of the respective genes[47]. On the other hand exportin-2 was found overexpressed in a large variety of malignancies[28] including HCC[48], correlated with poor patient outcome[49] and was identified as a p21-dependent p53 repression target in HCC[48]. These data together indicate a context-dependent dual role of regulatory circuits between the p53-p21 axis and nuclear transport proteins in HCC. We anticipate future studies to further define the multilayered crosstalk and feedback mechanisms between the nuclear trans-port machinery and the p53 network.

## Methods
**Analysis by peptide-labeled quantitative mass spectrometry.** Lysates isolated from nonsense-(All-Stars duplex, Qiagen, Hilden, Germany) and Nup155-siRNA transfected cells (72 h) treated with Nutlin-3a for 24 h were processed and analysed by LC-MS/MS. Two distinct siRNAs against Nup155 were used and for each one of

them a biological duplicate was analysed (four samples in total). For each condition harvested cells (PBS, concentration 106/100 µL) were lysed in urea buffer (10 M urea in 250 mMNH$_4$HCO$_3$) and 2% (w/v) Rapigest (Waters, Eschborn, Germany) reaching a final concentration of 4 M and 0.2% (w/v), followed by three sonication steps of 30 s each in a vial tweeter interrupted by a cool down on ice for 1 min. Reduction of solubilized proteins was achieved by using 10 mM DTT (at 37 °C for 30 min). The addition of 15 mM iodoacetamide (IAA) for 30 min. resulted in alkylation of cysteine rests. The protein digestion was performed by incubation with LysC (Wako Chemicals, Neuss, Germany) using a dilution of 1:100 (w/w) for 4 h at 37 °C followed by dilution of urea concentration to 1.5 M and addition of trypsin (Promega, Mannheim, Germany) to a dilution of 1:50 (w/w) overnight at 37 °C. After digestion samples were acidified with 10% (v/v) TFA. The cleavage by Rapigest was allowed to proceed at 37 °C for 30 min. Digested samples were clarified by centrifugation (17,000 × g for 5 min at RT) followed by a desalting step using C18 SEP-pack columns (Waters, Eschborn, Germany). During the desalting procedure, peptides were labeled by reductive dimethylation as described previously[50]. Controls were labeled using light formaldehyde (mass shift + 28) while Nup155-siRNA transfected samples were labeled using deuterated (heavy) formaldehyde (mass shift + 32). Labeling efficiency (>95%) was assessed by LC-MS/MS on an aliquot of each sample. Control and siRNA treated samples were then mixed 1:1 and subjected to off-line high-pH peptide fractionation. Thirty fractions were collected for each sample and pooled into 10 final fractions that were analysed by LC-MS/MS.

Fractionated peptides were analysed in a LTQ-Orbitrap Velos Pro (Thermo Fischer Scientific, Bremen, Germany) connected to a nanoAcquity UPLC system (Waters, Eschborn, Germany) equipped with a BEH300 C18 (75 µm × 250 mm, 1.7 µm) nanoAcquity UPLC column (Waters GmbH, Eschborn, Germany). For data acquisition in a data-dependent mode a TOP-20 strategy was used: survey MS scans (m/z range 375–1600) were acquired in the orbitrap (R = 30,000 FWHM) fragmenting up to twenty of the most abundant ions per full scan by collision-induced dissociation (normalized collision energy = 35, activation Q = 0.250) and analyzing them in the ion fall (LTQ). For full scans ion targets were 1,000,000 (or 500 ms maximum fill time) and 10,000 (or 50 ms maximum fill time) for MS/MS scans. Rejected were charge states "1" and "unknown". Repeat count = 1, exclusion duration = 60 s, list size = 500 and mass window ± 15 ppm enabled for dynamic exclusion.

Raw files were processed using MaxQuant v1.3.0.5[51] and searched against the human Swiss Prot entries of the Uniprot release 2012_14 with the Andromeda search engine[52] using the following search parameters: (1) Requirement of full tryptic specificity (cleavage after lysine or arginine residues, unless followed by proline); (2) Only two missed cleavages were allowed; (3) Carbamidomethylation (C) was used as fixed modification; (4) Variable modifications were oxidation (M) and acetylation (protein N-term); (5) 20 ppm (for precursor ions) and 0.5 Da (for fragment ions) were set for mass tolerance; (6) The option "Match between runs" was limited to a time window of 2 min. The reversed sequences of the target database served as decoy database. The false discovery rate was set to 1% at both peptide and protein level using target-decoy strategy[53]. For quantitative analysis, the MaxQuant output was processed using routines written in R (version 2.14.2). Only protein groups being identified by at least 2 razor + unique peptides, with a ratio count ≥2 and a ratio variability <100 were retained (3523 protein groups). In order to exclude off-target effects, the replicates from the two independent siRNAs were averaged and the R-package "fdrtool"[54] was used to fit a two components model on the averaged ratio distribution. Protein groups with a ratio belonging to the non-null component (q value < 0.1) were considered as affected by the depletion of Nup155.

**Cell culture.** The cell lines HepG2, SkHep1, and H1299 were purchased from ATCC (American Type Culture Collection, Manassas, USA) and HLE, HLF, and HuH7 were purchased from JCRB (Japanese Collection of Research Bioresources Cell Bank) and have been validated using STR-analyses (Leibniz-Institut, DSMZ, Braunschweig, Germany). All cell lines have regularly been tested for mycoplasma contamination using the MycoAlertTM Plus Mycoplasma Detection Kit (LT07, Lonza, Cologne, Germany). HepG2 cells were maintained in Roswell Park Memorial Institute 1640 medium (RPMI), SkHep1, HLE, HLF, H1299 and HuH7 cells were maintained in Dulbecco's Modified Eagle's Medium (DMEM), both media supplemented with 10% fetal calf serum (FCS) and 1% penicillin/streptomycin in an atmosphere containing 5% carbon dioxide. The Hep3B-4Bv (expressing the temperature-sensitive p53val135 mutation) were kindly provided by M. Oren[21] and maintained in DMEM with 10% FCS, 1% penicillin/streptomycin, and 1 mg/ml puromycin. H1299 cells and their derivative tet-off H24-p21-inducible cell line were kindly provided by C. Prives[22]. H24-p21-inducible cells received additionally doxycycline (5 mg/ml), puromycin (1 mg/ml), and G418 (100 mg/ml). p21 expression was induced by removing tetracycline antibiotics 72 h before harvesting the cells. RPMI, DMEM, and antibiotics were purchased from Sigma–Aldrich (Taufkirchen, Germany). Nutlin-3a (Sigma–Aldrich, Taufkirchen, Germany) was dissolved in DMSO (Calbiochem/Merck Biosciences, Schwalbach/Ts., Germany) and used in a final concentration of 10 µM for 24 h or 48 h as indicated.

**Transfection of siRNAs/cDNAs and cycloheximide experiments.** The siRNA transfections were performed with Oligofectamine (Invitrogen, Karlsruhe, Germany) according to the manufacturer´s instructions and were used at a final concentration of 50 nM. The transfected gene-specific siRNAs were purchased from Eurofins MWG Operon (Ebersberg, Germany) and QIAGEN (Hilden, Germany). The siRNA sequences are listed in Supplementary Table 3. As a negative control for all knockdown experiments served the QIAGEN All-Stars duplex.

Plasmids containing p21 full-length (FL) and p21 lacking the 3′UTR (Δ3′UTR) were kindly provided by C. Prives[16]. cDNAs encoding p21 lacking the 3′UTR and the 5′UTR (Δ3′Δ5′UTR) were amplified by PCR with a Platinum Pfx DNA polymerase (Invitrogen, Karlsruhe, Germany) using p21 Δ3′UTR as a template. The resulting p21 Δ3′Δ5′UTR cDNA was then cloned into a pcDNA3.1 vector (Invitrogen, Karlsruhe, Germany). p21 plasmid with alternative phenylalanine codons (p21_mut_Δ3′Δ5′UTR) was purchased from QIAGEN (Hilden, Germany). HDAC4-Flag pcDNA3 was a gift from Eric Verdin (Addgene plasmid # 13821;[55]). The gateway destination vector containing N-terminally His6-HA-StrepII-tagged Nup155[12] was provided by the Beck laboratory. Transient plasmid transfections were performed with Lipofectamin2000 (Invitrogen, Karlsruhe, Germany) according to the manufacturer´s protocol.

For protein half-life experiments transfected cells were treated with 50 µg/ml cycloheximide (C4859, Sigma–Aldrich, Taufkirchen, Germany) and harvested at the indicated time points.

**Gel electrophoresis and immunoblotting.** The harvesting of pretreated cells was performed by using a cell lysis buffer (Cell Signaling/New England Biolabs, Frankfurt, Germany) supplemented with a protease inhibitor cocktail (Serva, Heidelberg, Germany). Protein content of whole protein lysates were determined by Bradford assays (#500–0006, Protein Assay Dye Reagent Concentrate, Bio-Rad Laboratories GmbH, Munich, Germany). After boiling samples for 5 min at 90 °C, proteins were separated by SDS/PAGE and transferred to nitrocellulose membranes (Whatman, Dassel, Germany). Membranes where blocked with 5% milk/TBST (Milchpulver, Carl Roth GmbH und Co. KG, Karlsruhe, Germany) for 1 h and incubated overnight with the indicated antibodies diluted in blocking solution: anti-Nup155 rabbit polyclonal (dilution 1:500, #R34951, Atlas, Stockholm, Sweden); anti-Nup35 rabbit polyclonal (dilution 1:500, #R07940, Atlas, Stockholm, Sweden); anti-Nup93 rabbit polyclonal (dilution 1:500, #R07068, Atlas, Stockholm, Sweden); anti-Nup188 rabbit polyclonal (dilution 1:1000, NBP1–28748, Novus Biologicals, Littleton, US); anti-p21 rabbit polyclonal (dilution 1:250, sc-397, Santa Cruz, Heidelberg, Germany); anti-p21 mouse monoclonal (dilution 1:250, sc-6246, Santa Cruz, Heidelberg, Germany); anti-p53 mouse monoclonal (dilution 1:10000, #554293, Pharmingen/BD Transductional Laboratories, Heidelberg, Germany); anti-p53 rabbit polyclonal (dilution 1:500, sc-6243, Santa Cruz, Heidelberg, Germany); anti-mdm2 mouse monoclonal (dilution 1:250, sc-965, Santa Cruz, Heidelberg, Germany); anti-GFP mouse monoclonal (dilution 1:200, sc-9996, Santa Cruz, Heidelberg, Germany); anti-FTSJ1 rabbit polyclonal (dilution 1:500, PA5–35886, Thermo Scientific, Offenbach, Germany); anti-Drosha mouse monoclonal (dilution 1:200, sc-393591, Santa Cruz, Heidelberg, Germany); anti-Flag mouse monoclonal (dilution 1:3000, F1804, Sigma–Aldrich, Taufkirchen, Germany); anti-HA.11 mouse monoclonal (dilution 1:500, #MMS-101R, Covance, Princeton, NJ, USA), anti-HDAC4 (dilution 1:200, sc-365093, Santa Cruz, Heidelberg, Germany). The detection of ß-actin, monoclonal mouse anti-actin antibody in a dilution 1:10000 (#691001, MP Biomedicals, Illkirch, France) or vinculin, mouse monoclonal anti-vinculin antibody in a dilution 1:5000 (V9131, Sigma–Aldrich, Taufkirchen, Germany) verified an equal loading of each sample. After rinsing with TBST the membranes were incubated for 1 h with the corresponding fluorescent-secondary antibodies (LI-COR Bioscience, Bad Homburg, Germany). After rewashing with TBST detection was performed using Odyssey Sa Infrared Imaging System (LI-COR Bioscience, Bad Homburg, Germany). Alternatively, blots were incubated for 1 h with the corresponding secondary antibodies (1:2000 horseradish peroxidase goat anti-mouse ab6789 and horseradish peroxidase goat anti-rabbit ab6721, Abcam, Cambridge, UK) at room temperature. After rewashing with TBST visualization was performed by enhanced chemiluminescence (SuperSignal West Femto #34095, Thermo Scientific, Offenbach, Germany) using Bio-Rad universal hood II (Bio-Rad Laboratories GmbH, München, Germany). Densitometric quantification of immunoblots was performed using the Image Studio Software (v2.1.10, LI-COR Bioscience, Bad Homburg, Germany) or Quantity one 1-D Analysis Software (Bio-Rad Laboratories GmbH, München, Germany) including the normalization to the loading control actin. Uncropped scans of all blots shown in the figures are displayed in the Source Data file.

**Total RNA isolation, cDNA synthesis, and qRT-PCR.** Total cellular RNA was isolated by using the NucleoSpin RNA II kit (Macherey-Nagel, Düren, Germany). 1 µg RNA was reverse transcribed by using the Sigma–Aldrich (Taufkirchen, Germany) RT-PCR kit according to the manufacturer's instructions. Samples were analysed in duplicate or triplicate reactions on a StepOnePlus real-time PCR instrument (Applied Biosystems, Darmstadt, Germany) using the Absolute qPCR SYBR Green ROX Mix (Thermo Scientific, Offenbach, Germany). Alternatively, samples were analysed in duplicate or triplicate reactions on a Corbett Research Rotor-Gene 6000 real-time PCR instrument (Corbett Research Mortlake, NWS, Australia) using the SensiFAST SYBR No-Rox Mix (Bioline, London, UK). Expression levels were normalized to those of L32. Primers were designed with Primer-BLAST (NCBI) and obtained from APARA Bioscience GmbH (Denzlingen,

Germany) or from ThermoFisher Scientific (Offenbach, Germany). Primer sequences are listed in Supplementary Table 4.

**Co-immunoprecipitation.** HepG2 cells were co-transfected with plasmids expressing N-terminally HA-tagged Nup155 and C-terminally Flag-tagged HDAC4 using the FuGENE HD Transfection Reagent (Promega, Mannheim, Germany) and harvested 24 h later in non-denaturing lysis buffer. Dynabeads Protein G (Invitrogen, Karlsruhe, Germany) were incubated with anti-Flag (F1804, Sigma–Aldrich, Taufkirchen, Germany) antibody or IgG1 (Santa Cruz, Heidelberg, Germany) as negative control for 4 h. Following a washing step with PBS, beads were mixed with whole protein lysates and incubated overnight at 4 °C on an overhead rotator. Immunoprecipitated proteins were eluted by shaking of Dynabeads in 1x Laemmli buffer for 20 min at RT. Eluted Co-IP lysates were separated using SDS/PAGE and transferred to nitrocellulose membranes (Whatman, Dassel, Germany).

**Immunofluorescence staining.** Cultured cells were grown on glass coverslips and treated with the indicated siRNAs and drugs. Coverslips were rinsed repeatedly with PBS containing 2 mM MgCl2 at 37 °C and fixed with 4% paraformaldehyde in PBS for 10 min. Fixed cells were washed repeatedly with PBS for 20 min, permeabilized in 0,1 % Triton/PBS for 5 min and again rinsed with PBS for 10 min. Primary Antibody (MAb414, 1:2500) was diluted in antibody diluent (#S2022, Dako, Hamburg, Germany) and cells were incubated in a humid dark chamber. After 1 h cells were again rinsed with PBS for 10 min. Secondary antibody (Alexa Fluor 488 goat anti-mouse, dilution 1:200, Molecular Probes, Eugene, OR) was diluted in antibody diluent and coverslips were again incubated for 30 min. in a humid dark chamber. Cells were then washed with PBS for 10 min. and then rinsed in dH2O. Cells were mounted and counterstained with DAPI Fluoromount-G (0100–20, Southern Biotech/Biozol, Eching, Germany). Imaging was performed on a LSM 780 (Zeiss GmbH, Oberkochen, Germany) confocal microscope with a ×63 objective (Plan-Apochromat 63×/1.4 Oil Zeiss) using the 488 and 405 nm laser lines for fluorophore excitation.

**Immunocytochemical staining.** The pellets of HepG2 cells were mixed with agarose (#85011, Biozym Scientific GmbH, Hessisch Oldendorf, Germany) in a 1:1 ratio and transferred on parafilm (using a cut pipet tip) placed on ice to allow solidification for 30 min. The pellet was formalin fixed and paraffin embedded and 2 µm thick sections were prepared using a microtome. Sections were subjected to an automated immunohistochemical staining procedure using a Ventana Benchmark Ultra instrument (Roche Diagnostics AG, Rotkreuz, Switzerland). The mouse anti human p21 antibody (sc-6246, Santa Cruz, Heidelberg, Germany) was used as primary antibody (1:100 dilution, 32 min at 37 °C) with pH 9 pretreatment prior to peroxidase blocking.

**Cell fractionation.** Cells were treated with the indicated siRNAs and drugs. Cell fractionation was performed using the NE-PER Nuclear and Cytoplasmic Extraction Reagents (78835; ThermoFisher Scientific, Offenbach, Germany) according to the manufacturers' protocol.

**Human and murine HCC samples.** In this study the Affymetrix U133A2.0 gene expression dataset derived from 247 HCC patients was used that is publicly available through Gene Expression Omnibus accession number GSE14520 (http://www.ncbi.nlm.nih.gov/geo)[56,57]. The patient samples were obtained with informed consent from patients at the Liver Cancer Institute (LCI) and Zhongshan Hospital (Fudan University, Shanghai, China). Raw gene expression data were normalized using the robust multiarray average method and global median centering. The mean gene expression was calculated for genes with more than one probe set.

The laboratory of L. Zender kindly provided the samples of previously published murine HCC models[27], which have been approved by the committees of the regional authorities of the states of Lower Saxony (Niedersaechsisches Landesamt für Verbraucherschutz und Lebensmittelsicherheit; authorization number: M8/12) and Baden-Wuerttemberg (Regierungspraesidium Tuebingen; authorization numbers: M14/14, M04/15, M15/15).

**Polyribosome analysis.** siRNA transfections were performed as described above. 72 h post-transfection, 50 µg/ml cycloheximide was added to the growth media, cells were incubated for 30 min at 37 °C, placed on ice, washed 2x with ice-cold PBS containing 50 µg/ml cycloheximide, harvested by scraping with cell scraper and collected by brief centrifugation. The cell pellet was resuspended in lysis buffer (20 mM Tris-HCl at pH 7.5, 3 mM MgCl$_2$, 20 mM NaCl, 0.5% Triton X-100, 1% Tween-20, 1% sodium deoxycholate) supplemented with 0.1 U/ul RNAsin and 1x protease inhibitor without EDTA (Roche, Mannheim, Germany) and incubated on ice 10 min before centrifugation at $10,000 × g$ for 10 min at 4 °C. Cleared supernatants were loaded onto a 12 ml sucrose gradient (20 to 50% sucrose in 75 mM KCl, 1.5 mM MgCl2, 10 mM Tris-HCl at pH 7.5 containing 5 µg/ml cycloheximide). The gradients were centrifuged at 35,000 rpm for 2 h 39 min at 4 °C in an SW40Ti rotor (Beckman Coulter) with maximum acceleration and deceleration. Fractions of 1 ml or 0.5 ml were collected with continuous monitoring of UV254nm absorbance. RNA was extracted using TRIzol LS reagent (Invitrogen,

Karlsruhe, Germany) after an equal amount of in vitro-transcribed Renilla luciferase RNA was added to each fraction to correct for potential variation in extraction. The Maxima First Strand cDNA Synthesis kit (K1642, ThermoFisher Scientific, Offenbach, Germany) was used for reverse transcription (RT) and an in vitro-transcribed Firefly luciferase RNA was added as an internal standard to correct for potential variation in RT efficiency. Real-time PCR was performed as described above with expression levels normalized to Renilla spike-in RNA. Final number of gradients per condition result from initially performed ($n = 6$) and during the revision process performed gradients ($n = 10$): control (AS) siRNA ($n = 5$), Nup155 knockdown ($n = 7$; 5x siRNA#2 + 2x siRNA#3), FTSJ1 knockdown ($n = 4$; 2x siRNA#1 + 2x siRNA#2).

**Statistical analysis and software.** Data are presented as the mean of three independent experiments including standard deviation (stdv.) except otherwise specified. Statistical significance was evaluated by Student's t-test using Excel 2010 (Microsoft, Redmond, WA, USA). Expression differences in human liver cancer samples were assessed by nonparametric Mann–Whitney U-test using GraphPad Prism 6 (GraphPad Software, Inc, La Jolla, CA, USA). Box-plot elements are center line: median; box limits: upper and lower quartiles. Overall survival analysis in HCC patients was performed by Kaplan–Meier Curves and log-rank p values using GraphPad Prism 6. The statistical significance was defined as $p < 0.05$.

**Reporting summary.** Further information on research design is available in the Nature Research Reporting Summary linked to this article.

## Data availability

The mass spectrometry proteomics data have been deposited to the ProteomeXchangeConsortium (http://proteomecentral.proteomexchange.org) via the PRIDE partner repository[58] with the dataset identifier PXD012373. The source data underlying Figs. 1B-C, 2A–D, 3A-F, 4A-B, 5A-D, 6A-C, 7A and Supplementary Figs 1A-F, 2A,B,D, 3A-B, 4A,C-E, 5A-B, 6A-D are provided as a Source Data file. A reporting summary for this Article is available as a Supplementary Information file. All other data supporting the findings of this study are available from the corresponding author on reasonable request.

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

## Acknowledgements

K.H. received a fellowship by the Rahel-Goitein-Straus Program of the Medical Faculty Heidelberg and support by the Junior Academy of the German Society for Pathology. S.S was supported by DFG grant Si1487/3–1, TRR/SF209 (B04), by the Hella-Buehler-Foundation and by a HRCMM (Heidelberg Research Center for Molecular Medicine) Career Development Fellowship. S.R. and K.B. were supported by TRR/SF209 (B01) and (B03), respectively. P.S. and S.R. were supported by European Union's Horizon 2020 research and innovation program under grant agreement No 667273 (HEP-CAR). We thank Eva Maria Eiteneuer and Bianca Köhler for excellent technical assistance.

## Author contributions

K.H., E.D., A.C., A.D., A.A.P., P.R., M.T.M. and J.B. performed research. K.H., A.O., A.C. and S.R. analysed data. D.D. and L.Z. contributed murine HCC samples. S.S, P.S., J.G., M.B., K.B., E.H. and F.D. designed research. S.S, J.G., A.D., A.C. and M.B. wrote the manuscript.

## Additional information

**Competing interests:** The authors declare no competing interests.

