## [Peer Review File · Nature Communications]

Reviewers' comments:

Reviewer #1 (Remarks to the Author):

Specific comments

The authors focus on how the nucleotransport machine, embodied by Nup155 impacts on p53 pathway activation. The manuscript premise is sound; after depleting cells of Nup155 using targeted siRNA, the authors find that, of the > hundred genes induced by p53, there is one dominant Nup155-dependent effector when p53 is activated, p21. This unbiased screen is important, since the vast majority of new genes found to be induced by p53 or binding to p53 are usually not filtered with respect to penetrance. This study however, identified an outlier in an unbiased approach, and suggest more relevance to p53-p21 axis.

In defining a mechanism of how Nup155 impacts on p21, the authors further filter their outlier data, perform another siRNA screen and identify FTSJ1 as a cofactor. This unbiased approach further identified dominant cofactors in p21 control that were previously not identified. The data are novel.

General comments

The immunoblots visually reflect the conclusions. However, the immunoblotting data requires quantitation; eg figure 1, figure 3.

The supplementary data can be incorporated into the document (such as SF3). However, again most of these data (immunoblots) are now quantified.

As is practise in the mass spectrometry field, the raw data from the Orbitrap can be uploaded to Pride for community access and review access in the first instance.

Summary

The authors have identified two new regulators of p53-dependent induction of p21. This unbiased approach is relatively novel and highlights possible penetrance (importance) in p53 regulatory proteins that is often lacking. However, the function of FTSJ1 has not been defined. We cannot presume FTSJ1 functions through rRNA methylation since it might have an independent protein-interactions activity through which it functions. In addition, the function of Nup155 has not been defined; does it function through RNA translation or RNA export? However, the co-expression of Nup155 and FTSJ1 in cancers and their co-suppression by p53 is noted.

The title and narrative could be changed or clarified so as not to mislead readers since other interpretations are possible; p21 gene expression, RNA export, RNA loading to ribosomes, or RNA translation might be a (co-ordinated mechanism).

Reviewer #2 (Remarks to the Author):

The authors have performed large scale proteomics, polysome fractionation and a focused RNAi screen to study the putative role of the nucleoporin Nup155 in hepatocellular carcinoma (HCC). Based on their data they suggest Nup155 controls translation of the p53 target p21 via FTSJ1, a putative tRNA and rRNA methyltransferase. Furthermore, they identified Nup155 and FTSJ1 as repression targets of p53 in distinct HCC models, both mouse and human. While the results presented are potentially interesting, the manuscript substantially lacks quantitative analysis. The authors often claim significant changes without statistical analysis. For most experiments, it remains unclear whether the data are based on only the one Western blot that is presented or how often the experiment has been repeated. Loading controls are frequently overexposed so that it is difficult to judge if the amounts of proteins are in fact equal in the different lanes.

Specific comments:

1. Blots shown in Fig. 1B, 2A, 2C, 3A-C, 4A-B, S1, S2B, S3A, S4A need quantification from at least 3 independent experiments.

2. In Fig. 2D, the presented p-values for the polysome analysis are statistically not significant, whereas the authors claim the opposite in the results section (lines 189-195). Please clarify.

3. In line 168/169, the authors have written that Nup155 was previously reported to be involved in mRNA export. This is not exactly correct. In the referenced study by Zhang et al, it was shown that Nup155 is required for Hsp70 mRNA export, whereas bulk mRNA export was not studied. Furthermore, Nup155 depletion also affected Hsp70 nuclear import. With respect to the present study, Hsp70 mRNA export should be included as positive control to convincingly proof that p21 mRNA export is not affected by Nup155 depletion. And also in line with the Zhang study, p21 nuclear import should be studied in dependence on Nup155.

4. I assume that Fig. S2B shows blots from tet off extracts. Please specify.

5. Concerning the DROSHA depletion experiments: the authors claim that DROSHA depletion resulted in an increase of endogenous p21, whereas co-depletion of DROSHA and Nup155 decreased p21 increase in Nutlin-3a treated cells. The presented data do not support this notion. For Drosha#1 and Nup155#1 and 2 siRNA, p21 levels are indistinguishable as compared to the control. For Drosha#2 there might be a decrease, but it is hard to judge, as these lanes are at the edge of the gel and simply the transfer might not be optimal. Quantification is necessary and consistent results for the distinct siRNAs.

6. Please specify what H1299 cells are and why they were used.

7. When analysing the effect of the 5'UTR and the 3'UTR on p21 translation, the authors claim that both regions unexpectedly are not involved in the regulation of p21 mRNA translation (lines 215-220). Why unexpectedly? Please explain. Moreover, while I tend to agree that the 3'UTR in fact has not influence (given a proper quantification of the blots), depletion of both UTRs led to a complete loss of p21 expression, suggesting that the 5'UTR has an impact. Please clarify.

8. Next the authors claim that only depletion of FTSJ1, but not of other candidate factors involved in mRNA translation, could phenocopy the effect of Nup155 depletion on p21. However, in Fig. S3B, they show that for RPL7L and GTPBP4 at least one siRNA for each also led to a decrease in p21 protein levels. Please explain.

9. In line 241, the authors wrote "absence of Nup155", but I assume "absence of FTSJ1" is meant, as indicated in Fig. 3D and in the figure legend.

10. In Fig. 3C, the authors show that FTSJ1 depletion affects p21 protein levels, but not transcript levels in Nutlin-3 treated cells. Doesn't this show that the observed effects are p53-independent?

11. In line 249, the authors refer to Fig. 1B in which the effect of p53 induction of FTSJ1 has been studied. However, FTSJ1 levels haven't been analysed in this Figure.

12. Maybe it is helpful to use the same name for p53 throughout the manuscript or at least to introduce that Trp53 is used for the mouse protein/gene.

13. What are the exact p-values for the data presented in Fig. 4E? Given the large error bars, I would assume that they are at the very limit to significance.

14. The Discussion could be more focussed.

15. The entire manuscript needs editing.

Reviewer #3 (Remarks to the Author):

In this manuscript, the authors follow up on a study (Boyault et al., 2007) that identified a subgroup of hepatocellular carcinoma (HCC) characterized by mutation of TP53 and overexpression of Nup155, a structural nucleoporin. They aimed here at characterizing the link between the p53 pathway and Nup155.

Using a quantitative proteomic, the authors found that p21 levels are altered upon Nup155 siRNAs treatment followed by p53-induced activation. Having excluded other hypothesis, they found a mild decrease of p21 mRNA in the polysome fraction suggesting that Nup155 could be required for p21 mRNA translation.

At that stage, they went back to the initial proteomic study and searched for candidates reported to be involved in mRNA translation. Among five candidates analyzed using RNAi, FTSJ1 was found to

regulate p21 protein but not mRNA levels. The authors thus concluded that Nup155 positively regulates p21 mRNA translation through FTSJ1 and that this defect is likely independent on the possible FTSJ1 tRNA methyltransferase activity.

Finally, the authors show that Nup155 and FTSJ1 protein levels are negatively regulated by p53 induction and confirm in a cohort of HCC patients that Nup155 and FTSJ1 mRNA levels are increased in a correlated manner in tumor/non tumor tissues, and albeit to lower extent, in HCCs containing mutant p53 compared to tumors with wt p53.

Together, this study is well conducted, and, with the exception of the analysis of mRNA translation (see below), the data are clear and convincing. Yet the molecular mechanisms underlying the regulations between p53, Nup155, FTSJ1 and p21 remain to be defined.

1) Analyses of p21 mRNA translation

this is a critical experiment, as it provides a key message highlighted in the title : "translational control by Nup155..." or in the discussion "a link between a scaffold Nup and mRNA translation as shown here for Nup155...;"

Since other hypotheses were excluded (normal export and localization of the mRNAs, no significant changes of protein stability), altered traduction is indeed an expected mechanism, but the data as they are presented are not convincing:

in Figure 2D: the p-value is 0.0581 ($p > 0.05$ is to my knowledge usually considered as not significant). Yet, the author only indicate in the text (line 189):

"the amount of p21 mRNA in the subpolysomal fraction increased upon Nup155 knockdown as compared to siRNA control treatments ... Collectively,

these data suggest that Nup155 is required for p21 mRNA translation".

Although they indeed do not refer to a "significant increase", this is quite misleading and this trend should be more carefully validated.

- How often was this experiment repeated ($n=?$, arising from X independent polysome gradients). Providing as now requested by some journals, the values when $n < 4-5$ may help to understand this lack of significant variation.

- Instead of pooling all fractions, it would be more informative to provide the data independently for each of the polysome fraction.

- Because the authors suggest that Nup155 exerts its function via FTSJ1, the effect of FTSJ1 depletion on p21 translational control should be similarly analyzed using polysome fractionation.

- to strengthen this part, the two independent siRNAs should be used

- the method should be more carefully detailed in the corresponding section: are the various sub-polysome and polysome fractions mixed prior to RNA extraction, reverse transcription or RT-PCR ?

2) The author do not provide molecular mechanisms underlying each of these regulations, namely, they do not determine how:

(i) p53 negatively regulates Nup155 mRNA (and thus protein) levels:

what are the mechanisms? Transcription initiation, elongation, mRNA stability ...does this involve miRNA?...

(ii) Nup155 regulates FTSJ1 mRNA (and thus protein) level

as discussed (lines 344-353), several scenario are possible, but the possible link with HDAC4 might be tested (for instance, truncated mutant of Nup155 that cannot bind HDAC4)

(iii) FTSJ1 regulates p21 protein level, likely by regulating its translation.

Is this effect direct? Can the authors exclude that FTSJ1 regulates the translation of another target that would then in turn regulate p21?

In the discussion possible mechanisms are indeed suggested. To strengthen this study, at least one of these aspects could have been mechanistically analyzed.

3) in figure 3D: the author show that depletion of FTSJ1 leads to decreased level of p21 when cells are transfected with p21 Δ 3'UTR or p21_mut plasmids.

Since both constructs contain an intact 5'UTR, the p21 Δ 5' Δ 3' construct (used for Nup155 in Fig 3A) should also be tested.

4)mass-spectrometry data

- It would be important to perform a comparative mass-spectrometry analysis of cells depleted or not for FTSJ1 and then compare the list of affected targets with the one affected by Nup155. There were indeed a lot of blue dots in figure 1, that despite not being established p53 targets, could nevertheless be involved in this pathway.

- It is currently complex (if not impossible) to make the link between the raw data (tableS1) and the graphs of fig 1A and D. Therefore, the authors should

provide in table S1, a column with the intensity values (represented on the Fig1A_D in y-axis). This would for instance allow to determine which protein corresponds to the two red dots (p53-target) that are either more strongly affected or affected as p21 upon Nup155 RNAi (Fig 1A). For instance, are these genes also affected by FTSJ1 depletion ? are they regulated at the mRNA level? Related to

my previous point (3), one of these genes may provide the missing link between Nup155 and FTSJ1 or between FTSJ1 and p21...

- highlight in the table the genes referred to in figures 1A and 1D (i.e., list of all p53 targets, of all significant genes, of all Nups) and those used for the secondary RNAi screen (all genes involved in translation)

- the nomenclature used in this table differs from the one used in the text.... (for instance, a search for FTSJ1 does not enable to get the corresponding line since its short name here is TRM7-HUMAN - likewise, I could not identify C1ORF19, RPL71L or GTPBP4 that are listed in Figure S3... I assume they are named differently... Please modify the Table of the text so that the reader can have access to the raw data.

5) The effect of p53 depletion on Nup155 and FTSJ1 protein levels (Figure 4B) should be quantified (to provide an idea of the fold change) and analyzed also at mRNA level to allow subsequent comparison with the changes of RNA levels analyzed in the human patients (figure D, E)

6) Figure S3, please indicate the number in each categories.

Were only 5 candidates identified and assayed in this "mini-screen " ??

Provide the list of proteins involved in translation as defined here by the authors.

Here,

Response to referees

We thank the reviewers for their insightful and supportive comments. Based on the reviewers' suggestions we have conducted many additional experiments that have significantly strengthened the manuscript.

Below are our detailed responses to each comment of the three reviewers (copied in italics).

Reviewer #1 (Remarks to the Author):

Specific comments

The authors focus on how the nucleotransport machine, embodied by Nup155 impacts on p53 pathway activation. The manuscript premise is sound; after depleting cells of Nup155 using targeted siRNA, the authors find that, of the > hundred genes induced by p53, there is one dominant Nup155-dependent effector when p53 is activated, p21. This unbiased screen is important, since the vast majority of new genes found to be induced by p53 or binding to p53 are usually not filtered with respect to penetrance. This study however, identified an outlier in an unbiased approach, and suggest more relevance to p53-p21 axis.

In defining a mechanism of how Nup155 impacts on p21, the authors further filter their outlier data, perform another siRNA screen and identify FTSJ1 as a cofactor. This unbiased approach further identified dominant cofactors in p21 control that were previously not identified. The data are novel.

General comments

The immunoblots visually reflect the conclusions. However, the immunoblotting data requires quantitation; eg figure 1, figure 3. The supplementary data can be incorporated into the document (such as SF3). However, again most of these data (immunoblots) are now quantified.

We thank the reviewer for the supportive notes. In order to address the concern we have now performed densitometric quantification of the immunoblots shown in Figures 1B (mod.), 1C; 2A, 2E-H; 3B, 3C, 3F; 4A; S1E; S2F; S4A, S4B, S4C (mod.), S4D. The data including the level of statistical significance are shown in the corresponding main or supplementary figures and support our major conclusions (see also Table below (reviewer #2, comment 1.)). Additionally, according to the reviewers' suggestion we have incorporated the focused RNAi approach (formerly Fig. S3A) into the main Figure 3 (Fig. 3A). Unfortunately, we were not able to put all data shown in Figure S3 into the main Figures due to space limitations.

As is practise in the mass spectrometry field, the raw data from the Orbitrap can be uploaded to Pride for community access and review access in the first instance.

The raw mass spectrometry data has now been uploaded on PRIDE to allow community access as suggested by the reviewer, which is now mentioned in the Methods section. The submission details including the account details to allow access by the reviewer are as follows:

Project Name: Nuclear pore component Nup155 is part of the p53 network in liver cancer

Project accession: PXD012373

Project DOI: Not applicable

Reviewer account details:

Username: reviewer42488@ebi.ac.uk

Password: 5y2DcD9h

Summary

The authors have identified two new regulators of p53-dependent induction of p21. This unbiased approach is relatively novel and highlights possible penetrance (importance) in p53 regulatory proteins that is often lacking. However, the function of FTSJ1 has not been defined. We cannot presume FTSJ1 functions through rRNA methylation since it might have an independent protein-interactions activity through which it functions. In addition, the function of Nup155 has not been defined; does it function through RNA translation or RNA export? However, the co-expression of Nup155 and FTSJ1 in cancers and their co-suppression by p53 is noted.

The reviewer raises important points regarding the mechanistic links between Nup155, FTSJ1 and p21 mRNA translation. After performing additional experiments we can now show that Nup155 regulates FTSJ1 at the level of transcription via HDAC4 which interacts with Nup155 (Fig. 3I and Fig. S3F). Moreover, based on further polyribosome experiments we can demonstrate a translation defect for p21 mRNA, reflected by reduced ribosomal occupancy in the Nup155 and FTSJ1 knockdown conditions. We agree with the reviewer that we cannot prove that FTSJ1 functions through rRNA methylation in regulating p21 mRNA translation. Definitively, yet to be defined FTSJ1 interacting proteins could play a role in this context, as pointed out by the reviewer. Thus, we

carefully rephrased the respective sentence in the Results section as follows: "We thus conclude that Nup155 regulates p21 mRNA translation through FTSJ1, independent of the TTC codon usage and (besides other conceivable mechanisms including protein-protein interactions) possibly by its putative role in ribosomal RNA methylation." We also included this consideration in the respective paragraph of the Discussion section.

The title and narrative could be changed or clarified so as not to mislead readers since other interpretations are possible; p21 gene expression, RNA export, RNA loading to ribosomes, or RNA translation might be a (co-ordinated mechanism).

We believe that the additional data incorporated in the revised version of the manuscript offers more mechanistic insight linking p53, p21, Nup155 and FTSJ1. Nevertheless, we are aware of the limitation of the study and chose a more general title of the manuscript: „Nucleoporin Nup155 is part of the p53 network in liver cancer“

Reviewer #2 (Remarks to the Author):

The authors have performed large scale proteomics, polysome fractionation and a focused RNAi screen to study the putative role of the nucleoporin Nup155 in hepatocellular carcinoma (HCC). Based on their data they suggest Nup155 controls translation of the p53 target p21 via FTSJ1, a putative tRNA and rRNA methyltransferase. Furthermore, they identified Nup155 and FTSJ1 as repression targets of p53 in distinct HCC models, both mouse and human. While the results presented are potentially interesting, the manuscript substantially lacks quantitative analysis. The authors often claim significant changes without statistical analysis. For most experiments, it remains unclear whether the data are based on only the one Western blot that is presented or how often the experiment has been repeated. Loading controls are frequently overexposed so that it is difficult to judge if the amounts of proteins are in fact equal in the different lanes.

We want to thank the reviewer for pointing this out. This concern has also been raised by reviewer #1 and #3. We have made an effort to improve the quantitative analysis of our data, as explained in detail above and below.

Specific comments:

1. *Blots shown in Fig. 1B, 2A, 2C, 3A-C, 4A-B, S1, S2B, S3A, S4A need quantification from at least 3 independent experiments.*

We appreciate reviewers' concerns and performed densitometric analyses of the above mentioned immunoblots (derived from at least 3 independent experiments). These quantifications are illustrated as described by the following table:

initial	current	quantification
Fig 1B	Fig 1B (mod.)	Fig 1B, S1A
Fig 2A	Fig 2A	S2A
Fig 2C	Fig 2C	Fig 2C lower panel
Fig 3A	Fig 2E-H (G mod.)	Fig 2E-H lower panels
Fig 3B	Fig 3B	S3A
Fig 3C	Fig 3C	Fig 3C right panel, S3B
Fig 4A	Fig 4A	Fig 4A middle panel
Fig 4B	Fig S4A	Fig S4A
Fig S1	Fig S1E	Fig S1F
Fig S2B	Fig S2D	Fig S2D right panel
Fig S3A	Fig S2F (mod.)	Fig S2F middle and right panel
Fig S4A	Fig S4B-D (C mod.)	Fig S4B-D lower panels

2. *In Fig. 2D, the presented p-values for the polysome analysis are statistically not significant, whereas the authors claim the opposite in the results section (lines 189-195). Please clarify.*

We agree with the reviewer that the initial polyribosome analyses have only been close to statistical significance (p = 0.056) due to a limited number of experiments. We have now performed further polysome fractionation

experiments (10 additional polysome gradients) including treatments with a second Nup155 siRNA and two different FTSJ1 siRNAs with resulting significant p-values for the respective combined knockdown conditions ($p = 0.0015$ for Nup155 and $p = 0.016$ for FTSJ1 depletion). The data are now illustrated in Fig. 2D, S2E and Fig. 3E, S3C and strengthen one of the major points of the manuscript showing that Nup155 and FTSJ1 are both involved in p21 mRNA translation. As we combined data derived from gradients from the initial analyses as well as from gradients prepared during the revision process we ended up with overlapping control samples in Fig. 2D and Fig. 3E. This is noted in the corresponding figure legends.

3. In line 168/169, the authors have written that Nup155 was previously reported to be involved in mRNA export. This is not exactly correct. In the referenced study by Zhang et al, it was shown that Nup155 is required for Hsp70 mRNA export, whereas bulk mRNA export was not studied. Furthermore, Nup155 depletion also affected Hsp70 nuclear import. With respect to the present study, Hsp70 mRNA export should be included as positive control to convincingly prove that p21 mRNA export is not affected by Nup155 depletion. And also in line with the Zhang study, p21 nuclear import should be studied in dependence on Nup155.

We acknowledge and amend the unprecise phrasing. We intended to refer to the published role of Nup155 in HSP70mRNA export to support the consideration that Nup155 may also have a function in mediating the export of another mRNA such as p21 mRNA. It was not our intention to claim a documented role of Nup155 in bulk mRNA export. Neither did we intend to ignore a function of Nup155 in nuclear import of HSP70 protein. We have changed the phrasing in the respective Results paragraph accordingly: "Nup155 was previously reported to be involved in HSP70 mRNA export and HSP70 protein import."

Based on the reviewers' suggestion we tested if there is a shift between the nuclear and cytoplasmic distribution of HSP70mRNA. In fact, we could show that there was a significant decrease of the cytoplasmic/nuclear ratio for HSP70mRNA upon Nup155 depletion consistent with previously published data (Fig. S2B, right panel). However, we did not observe significant change of the cytoplasmic/nuclear ratio for p21 mRNA under the same conditions (Fig. S2B, left panel). We also determined if there is a nuclear import defect of p21 protein by immunocytochemistry of HepG2 cells after Nup155 siRNA treatment, which was also not detectable (Fig. S2C). Due to the rather small protein size of p21 (21kDa) we expect this protein to freely diffuse between the nucleus and cytoplasm independent of specific Nucleoporin or nuclear transport receptor interactions.

4. I assume that Fig. S2B shows blots from tet off extracts. Please specify.

The assumption of the reviewer is correct. As written in the manuscript the immunoblots show cycloheximide chase experiments in the tet-inducible p21 cells under tet-off conditions. We have corrected this labelling error in the current Fig. S2D (formerly Fig. S2B), accordingly.

5. Concerning the DROSHA depletion experiments: the authors claim that DROSHA depletion resulted in an increase of endogenous p21, whereas co-depletion of DROSHA and Nup155 decreased p21 increase in Nutlin-3a treated cells. The presented data do not support this notion. For Drosha#1 and Nup155#1 and 2 siRNA, p21 levels are indistinguishable as compared to the control. For Drosha#2 there might be a decrease, but it is hard to judge, as these lanes are at the edge of the gel and simply the transfer might not be optimal. Quantification is necessary and consistent results for the distinct siRNAs.

We agree with the reviewer that for Drosha#1 and Nup155#1 and #2 siRNAs, p21 levels were indistinguishable as compared to the control (formerly Fig. S3A). However, we do not consider this as a true rescue since the reference in this case should not be the AS control, but the Drosha knockdowns alone (lane 2 and 3) that show higher p21 levels to start with. We have included densitometric analyses of immunoblots from three independent experiments (Fig. S2F) showing that normalized to the corresponding Drosha knockdown conditions p21 protein is still significantly decreased by Nup155 depletion and therefore not rescued. Our interpretation of the data is still that in a scenario where miRNA biogenesis is largely blocked p21 protein remains sensitive to Nup155 knockdown indicating a miRNA-independent regulation of p21 by Nup155.

6. Please specify what H1299 cells are and why they were used.

H1299 is the parental lung adenocarcinoma cell line that the p21 tet-inducible cells (H24-p21) were derived from. These are very robust cancer cells that tolerate well co-transfection of siRNAs and cDNA expression constructs. Therefore, we used these cells for follow-up experiments based on the data derived from the p21 tet-inducible cells. Even though these carcinoma cells are of lung and not of liver origin, we believe that they are still appropriate for the conducted experiments as exogenously expressed p21 in the p21 tet-inducible cells or the parental H1299 show the same dependency on Nup155 as does endogenous p21 in HepG2 cells. Moreover, the data together may suggest that the observed effect could be transferred to other tumor entities. We have included in the relationship between H1299 and the p21-tet-inducible cells in the Material and Methods section.

7. When analysing the effect of the 5'UTR and the 3'UTR on p21 translation, the authors claim that both regions unexpectedly are not involved in the regulation of p21 mRNA translation (lines 215-220). Why unexpectedly? Please explain. Moreover, while I tend to agree that the 3'UTR in fact has not influence (given a proper

quantification of the blots), depletion of both UTRs led to a complete loss of p21 expression, suggesting that the 5'UTR has an impact. Please clarify.

We used the term "unexpectedly" because there are several factors that have been reported to control p21 mRNA translation such as RNA binding proteins that associate either with the 5'UTR such as CUG-binding protein (CUGBP1) and Calreticulin or proteins (e.g. Musashi) that bind to the 3'UTR of p21 mRNA, besides a couple of miRNAs that also bind to the 3'UTR. Therefore, we expected either the 3'UTR or the 5'UTR to play role in Nup155-dependent regulation of p21 mRNA translation. We mention this context in the Introduction and in the respective paragraph of the Results section. However, as this is apparently not obvious for the reader, we removed the term "unexpectedly" in the respective sentence.

Densitometric quantification of immunoblots of three independent experiments using the Δ 3'UTR p21 construct and the combined Δ 3' Δ 5'UTR p21 construct do not indicate a difference of the effect size in terms of reduction of p21 protein (current Fig. 2 E-H, quantification in lower panels). To avoid this impression we now show the p21 immunoblot in Fig. 2G (formerly Fig. 3A) in a slightly longer exposure. The experiments were conducted to test the hypothesis that removing either of the UTRs would lead to a rescue of the observed phenotype and from this perspective none of the UTR deletion constructs revealed an appropriate result.

8. Next the authors claim that only depletion of FTSJ1, but not of other candidate factors involved in mRNA translation, could phenocopy the effect of Nup155 depletion on p21. However, in Fig. S3B, they show that for RPL7L and GTPBP4 at least one siRNA for each also led to a decrease in p21 protein levels. Please explain.

We agree with the reviewer that one siRNA of RPL7L (RL7L) and GTPBP4 (NOG1) as shown in the current Fig. 3A) do reduce p21 protein level. However, p53 levels of the corresponding conditions do also decrease so that the phenotype of Nup155 knockdown (reduced p21 and unchanged p53 protein) is not phenocopied.

9. In line 241, the authors wrote "absence of Nup155", but I assume "absence of FTSJ1" is meant, as indicated in Fig. 3D and in the figure legend.

We thank the reviewer for bringing this to our attention. The correct phrase is indeed "absence of FTSJ1" which we changed accordingly.

10. In Fig. 3C, the authors show that FTSJ1 depletion affects p21 protein levels, but not transcript levels in Nutlin-3 treated cells. Doesn't this show that the observed effects are p53-independent?

We agree with the reviewer that the effect of Nup155 on p21 mRNA translation appears in the narrower sense independent of p53. However, p53 is the most important inducer of p21 and p53 does also repress Nup155 and FTSJ1 in a p21-dependent manner, based on the additional experimental data, so that the cross-talk between the aforementioned players is still embedded in the p53 network. Nevertheless, Nup155-dependent regulation of p21 mRNA translation may also apply to scenarios where p21 is induced by p53-independent mechanisms.

11. In line 249, the authors refer to Fig. 1B in which the effect of p53 induction of FTSJ1 has been studied. However, FTSJ1 levels haven't been analysed in this Figure.

As noted correctly by the reviewer, the phrasing was not precise. Therefore, we modified the sentence as follows: "As another important aspect of the crosstalk between Nup155, FTSJ1, and the p53 pathway we noticed that Nup155 and FTSJ1 proteins are reduced after p53 induction (Fig. 3B (for Nup155), and Fig. 3C (for FTSJ1), lane 1 vs. lane 4, respectively)."

12. Maybe it is helpful to use the same name for p53 throughout the manuscript or at least to introduce that Trp53 is used for the mouse protein/gene.

Based on the reviewers' suggestion we introduced Trp53 as the mouse gene/protein of human p53. The respective sentence in the Results section is therefore phrased as follows: "To test this hypothesis *in vivo*, we analyzed murine HCCs generated by transposon-based gene transfer of Myc and N-ras^{G12V} into mice with homozygous deletion of Trp53 (Trp53^{-/-}), the murine homolog of human p53, or into wild-type mice (Trp53^{+/+})"

13. What are the exact p-values for the data presented in Fig. 4E? Given the large error bars, I would assume that they are at the very limit to significance.

The p-values in former Figure 4E (current Fig. S4F) are as follows for Nup155 p = 0.048 and for FTSJ1 p = 0.023. We agree with the reviewer that the differences are rather mild and comment on this explicitly in the Discussion section. We believe that the differences are only mild due to the fact that HCCs of this cohort albeit being genetically p53 wild-type tumors are functionally p53 inactivated (e.g. by the HBX protein or other mechanisms). Therefore, the biological relevance of p53-mediated repression of both factors is probably best reflected in the (strongly) significant difference when comparing non-tumorous (functional p53 wt) with tumorous tissue (functionally or genetically inactivated p53) (Fig. 4E).

14. The Discussion could be more focussed.

In the light of the additionally performed experiments and acquired data we have shortened the Discussion section and focused on three major points that we consider important: a) NPC components involved in regulating translation and transcription, b) possible mechanisms participating in FTSJ1-dependent regulation of p21 mRNA, c) putting the p53/21-Nup155/FTSJ1 regulatory loop in the context of the HCC data and previously published observations. We hope that the current version is better appreciated by the reviewer.

15. *The entire manuscript needs editing.*

We are grateful for the notes and close attention of the reviewer, which have enabled us to further improve the manuscript and to address inconsistencies. We have carefully edited the manuscript to correct labelling and other errors. We hope that the manuscript in its current version meets the expectations of the reviewer in this regard.

Reviewer #3 (Remarks to the Author):

In this manuscript, the authors follow up on a study (Boyault et al., 2007) that identified a subgroup of hepatocellular carcinoma (HCC) characterized by mutation of TP53 and overexpression of Nup155, a structural nucleoporin. They aimed here at characterizing the link between the p53 pathway and Nup155.

Using a quantitative proteomic, the authors found that p21 levels are altered upon Nup155 siRNAs treatment followed by p53-induced activation. Having excluded other hypothesis, they found a mild decrease of p21 mRNA in the polysome fraction suggesting that Nup155 could be required for p21 mRNA translation. At that stage, they went back to the initial proteomic study and searched for candidates reported to be involved in mRNA translation. Among five candidates analyzed using RNAi, FTSJ1 was found to regulate p21 protein but not mRNA levels. The authors thus concluded that Nup155 positively regulates p21 mRNA translation through FTSJ1 and that this defect is likely independent on the possible FTSJ1 tRNA methyltransferase activity. Finally, the authors show that Nup155 and FTSJ1 protein levels are negatively regulated by p53 induction and confirm in a cohort of HCC patients that Nup155 and FTSJ1 mRNA levels are increased in a correlated manner in tumor/non tumor tissues, and albeit to lower extent, in HCCs containing mutant p53 compared to tumors with wt p53.

Together, this study is well conducted, and, with the exception of the analysis of mRNA translation (see below), the data are clear and convincing. Yet the molecular mechanisms underlying the regulations between p53, Nup155, FTSJ1 and p21 remain to be defined.

1) *Analyses of p21 mRNA translation*

this is a critical experiment, as it provides a key message highlighted in the title : "translational control by Nup155..." or in the discussion "a link between a scaffold Nup and mRNA translation as shown here for Nup155..."

Since other hypotheses were excluded (normal export and localization of the mRNAs, no significant changes of protein stability), altered traduction is indeed an expected mechanism, but the data as they are presented are not convincing:

in Figure 2D: the p-value is 0.0581 (p >0.05 is to my knowledge usually considered as not significant). Yet, the author only indicate in the text (line 189):

"the amount of p21 mRNA in the subpolysomal fraction increased upon Nup155 knockdown as compared to siRNA control treatments ... Collectively, these data suggest that Nup155 is required for p21 mRNA translation".

Although they indeed do not refer to a "significant increase", this is quite misleading and this trend should be more carefully validated.

- How often was this experiment repeated (n=?, arising from X independent polysome gradients). Providing as now requested by some journals, the values when n<4-5 may help to understand this lack of significant variation.

We agree with the reviewer that from a statistical perspective a p-value of 0.056 cannot be considered as significant and should be considered as a trend. Therefore, we initially avoided the term "significant" as the reviewer noted correctly. We also agree that the involvement of Nup155 in p21 mRNA translation is one of the key findings of the manuscript requiring conclusive data with statistical significance. For this reason, we performed additional polysome gradients (see below and also reply to Reviewer#2, 2nd comment). The resulting p-values for p21 mRNA comparing polysomal and subpolysomal fractions of the control siRNA and respective knockdown conditions are now below 0.05 (p = 0.0015 for Nup155 and p = 0.016 for FTSJ1 depletion) justifying the term "significant" which is now used. The previous data was derived from 6 polyribosome gradients (3 x AS control and 3 x Nup155 siRNA#2). The newly performed gradients included the following conditions: 2 x ctrl siRNA, 2 x Nup155 siRNA#2, 2 x Nup155 siRNA#3, 2 x FTSJ1 siRNA#1 and 2 x FTSJ1 siRNA#2. Therefore, we end up with the following final numbers per condition: 5 x control, 7 x Nup155 combined knockdown, 4 x FTSJ1 combined knockdown. This is also stated in the Methods section.

- Instead of pooling all fractions, it would be more informative to provide the data independently for each of the polysome fraction.

While we agree that, in principle, looking at each individual fraction more precisely determines the extent to which p21 mRNA exhibits lower ribosome occupancy in the knock downs vs. controls. However, we choose to pool fractions for two reasons. First, p21 mRNA has a short ORF (~500 bps) and therefore, even when translated at maximal efficiency, is largely associated with "light" polysome fractions. The second reason is technical: it is difficult to obtain enough RNA for use in downstream applications (i.e. reverse transcription reactions). We typically only get 120 ng/ul from pooling 8-9 fractions. This already pushes our limits and we emphasise that the key conclusion "Nup155 and FTSJ1 are both required for p21 mRNA translation" is strongly supported by the data derived from the pooled fractions.

- Because the authors suggest that Nup155 exerts its function via FTSJ1, the effect of FTSJ1 depletion on p21 translational control should be similarly analyzed using polysome fractionation.

Based on the reviewers' suggestion we also performed polyribosome gradients upon FTSJ1 knockdown (also discussed above). Supporting our hypothesis, we observed a significant increase of the subpolysomal and decrease of the polysomal p21 mRNA upon FTSJ1 depletion ($p = 0.016$), indicating a translation defect similar to the Nup155 knockdown condition (Fig. 3E and Fig. S3C).

- to strengthen this part, the two independent siRNAs should be used

We have now included two different siRNAs for the FTSJ1 and Nup155 knockdown conditions. See Fig. 2D and S2E for Nup155 and Fig. 3E and S3C for FTSJ1.

- the method should be more carefully detailed in the corresponding section: are the various sub-polysome and polysome fractions mixed prior to RNA extraction, reverse transcription or RT-PCR ?

The fractions were mixed prior to RNA extraction. We have added this information in the Results section: "Following separation by sucrose gradient, polysomal and subpolysomal mRNA fractions were pooled prior to RNA extraction and analyzed by qRT-PCR to assess the levels of translationally active versus inactive transcripts, respectively."

2) The author do not provide molecular mechanisms underlying each of these regulations, namely, they do not determine how:

(i) p53 negatively regulates Nup155 mRNA (and thus protein) levels:

what are the mechanisms? Transcription initiation, elongation, mRNA stability ...does this involve miRNA?...

(ii) Nup155 regulates FTSJ1 mRNA (and thus protein) level

as discussed (lines 344-353), several scenarios are possible, but the possible link with HDAC4 might be tested (for instance, truncated mutant of Nup155 that cannot bind HDAC4)

(iii) FTSJ1 regulates p21 protein level, likely by regulating its translation.

Is this effect direct? Can the authors exclude that FTSJ1 regulates the translation of another target that would then in turn regulate p21?

In the discussion possible mechanisms are indeed suggested. To strengthen this study, at least one of these aspects could have been mechanistically analyzed.

(i) As pointed out by the reviewer, "how p53 negatively regulates Nup155" is definitively an interesting question which we aimed to address in more detail. For further insight in the underlying mechanism, we first tested if decreased Nup155 protein is paralleled by a decrease in nascent (unspliced) Nup155 mRNA. As now shown in Fig 4A (lower panel) we could demonstrate that nascent Nup155 mRNA is similarly reduced upon p53 activation and that this reduction can be completely rescued by depletion of p53 (Fig. 4B and S4A). These findings suggest a negative regulation of Nup155 by p53 at the level of transcription. Repression targets of p53 are commonly negatively regulated at the transcriptional level via p21. To this end we tested if p53 mediated repression of Nup155 can be rescued by p21 depletion using two different siRNAs. In fact, reduced Nup155 transcript levels upon p53 activation could be reverted by silencing of p21 (Fig. 4C). Taken together, these data suggest that p53 represses Nup155 transcription in a p21-dependent manner and thus indicates a negative feedback loop.

(ii) We further agree with the reviewer that more insight into the mechanism by which Nup155 regulates FTSJ1 would add valuable information to the manuscript. Therefore, we used a similar approach as described above in (i). We first tested if the reduction in FTSJ1 mRNA (mature) upon Nup155 depletion, is associated with reduced nascent (unspliced) FTSJ1 mRNA, which was detected by intron-exon spanning primers. Indeed, nascent FTSJ1 mRNA was reduced upon Nup155 silencing (see Figure 3G). Thus, we assumed that Nup155 regulates either directly or indirectly FTSJ1 transcription. As noted by the reviewer and based on the work of Kehat et al (Ref. 24), HDAC4 could be a potential candidate in this scenario as it was shown to interact with Nup155. Indeed, we could recapitulate the interaction of Nup155 and HDAC4 by co-immunoprecipitation in HepG2 cells (Fig. S3E). Moreover, knockdown of HDAC4 resulted in decreased FTSJ1 nascent levels (Fig. 3H) while simultaneous overexpression of Nup155 and HDAC4 led to increased FTSJ1 transcription (Fig. 3I and Fig. S3F). These new data indicate that Nup155 regulates FTSJ1 transcription through its interaction partner HDAC4.

(iii) Due to the additionally performed polysome fractionation experiments we are able to show that FTSJ1 is involved p21 mRNA translation (Fig. 3E and Fig. S3C). However, (see also reply to Reviewer#1 below the summary) we agree with the reviewer that we cannot exclude that this regulation is indirect via (yet to be defined) interacting proteins, proteins that itself require FTSJ1 for their translation or by its function as a putative rRNA methyltransferase. Therefore, we have included these caveats in the Discussion section.

3) in figure 3D: the author show that depletion of FTSJ1 leads to decreased level of p21 when cells are transfected with p21 Δ 3'UTR or p21_mut plasmids. Since both constructs contain an intact 5'UTR, the p21 Δ 5' Δ 3' construct (used for Nup155 in Fig 3A) should also be tested.

We apologize for the incorrect labelling of the constructs. The constructs with the altered codon usage (p21_mut) were derived from the p21 Δ 3' Δ 5'UTR construct and as such do not contain the 5'UTR. Therefore, the data shown (Fig. 3F) are actually what the reviewer was asking for. We have fixed this in the figure to correctly represent the data.

4) mass-spectrometry data

- It would be important to perform a comparative mass-spectrometry analysis of cells depleted or not for FTSJ1 and then compare the list of affected targets with the one affected by Nup155. There were indeed a lot of blue dots in figure 1, that despite not being established p53 targets, could nevertheless be involved in this pathway.

We agree with the reviewer that, in principle, a comparative MS analysis upon FTSJ1 knockdown could be interesting and may help to identify additional players in the given context. However, we were greatly discouraged regarding this approach as we found out that the two other p53 targets PYCARD and CathepsinD are downregulated at the transcript level upon Nup155 knockdown and were not reduced at the protein level by FTSJ1 depletion (see also Figure and comment below). Thus, we believe that focusing on Nup155- and FTSJ1-dependent regulation of proteins/genes beyond the p53 network would be a worthwhile topic for a separate study but is beyond the scope of this study.

- It is currently complex (if not impossible) to make the link between the raw data (tableS1) and the graphs of fig 1A and D. Therefore, the authors should provide in table S1, a column with the intensity values (represented on the Fig1A_D in y-axis). This would for instance allow to determine which protein corresponds to the two red dots (p53-target) that are either more strongly affected or affected as p21 upon Nup155 RNAi (Fig 1A). For instance, are these genes also affected by FTSJ1 depletion? are they regulated at the mRNA level? Related to my previous point (3), one of these genes may provide the missing link between Nup155 and FTSJ1 or between FTSJ1 and p21...

We followed the suggestion of the reviewer and provide now two supplementary Tables with Table S2 showing all p53 targets (corresponding to the red dots in Figure 1A) and Table S3 showing all NPC components (corresponding to the purple dots in Figure 1A). The two red dots that the reviewer specifically refers to represent PYCARD and CathepsinD (see also comment above). As proposed by the reviewer we tested if the induction of both proteins after Nutlin-3a treatment is reduced by FTSJ1 knockdown in HepG2 cells. The Figure below illustrates that FTSJ1 knockdown did not significantly affect the induction of PYCARD or CathepsinD (A). Moreover, we found that both p53 targets were significantly less induced at the transcript level upon Nup155 siRNA treatment as measured by qRT-PCR (B). These data indicate a regulation of PYCARD and CathepsinD independent of FTSJ1. These data can be included in the manuscript upon reviewer recommendation.

A

B

A) Left panel: HepG2 cells were either treated with control siRNA (AS) or two FTSJ1 siRNAs (FTSJ1#1 and FTSJ1#2) for 72h. Nutlin-3a was added 24 h before harvesting. Cell extracts were analyzed by immunoblotting with the indicated antibodies. Middle and right panel: corresponding densitometric analyses **B)** HepG2 cells were either treated with control siRNA (AS) or two Nup155 siRNAs (Nup155#1 and Nup155#2) for 72h. Nutlin-3a was added 24 h before harvesting. Relative mRNA levels of Nup155 (left panel), CathepsinD (middle panel) and PYCARD (right panel) were measured by qRT-PCR.

- highlight in the table the genes referred to in figures 1A and 1D (i.e., list of all p53 targets, of all significant genes, of all Nups) and those used for the secondary RNAi screen (all genes involved in translation)

In addition to the supplementary tables S2 and S3 mentioned above, we also highlighted with the corresponding colors (red = p53 targets, purple = NPC components, green = p21, orange = Nup155, and yellow = candidates for the focused RNAi approach) the respective groups/proteins in the Table S1 (containing all quantified proteins).

- the nomenclature used in this table differs from the one used in the text... (for instance, a search for FTSJ1 does not enable to get the corresponding line since its short name here is TRM7-HUMAN - likewise, I could not identify C1ORF19, RPL71L or GTPBP4 that are listed in Figure S3... I assume they are named differently... Please modify the Table of the text so that the reader can have access to the raw data.

We have addressed this inconsistency using different aliases for the same proteins in Table S1 and the immunoblot of the focused RNAi approach (Fig. 3A). The final nomenclature (bold) is as follows

Sen15 (tRNA-splicing endonuclease subunit Sen15) = TSEN15, C1orf19
RL7L1 (60S ribosomal protein L7-like 1) = RPL7L1
NOG1 (Nucleolar GTP-binding protein 1) = GTP binding protein 4/GTBP4

5) The effect of p53 depletion on Nup155 and FTSJ1 protein levels (Figure 4B) should be quantified (to provide an idea of the fold change) and analyzed also at mRNA level to allow subsequent comparison with the changes of RNA levels analyzed in the human patients (figure D, E)

We followed the reviewers' suggestion and have quantified the protein and mRNA levels of Nup155 and FTSJ1 upon p53 depletion by densitometric analyses and qRT-PCR, respectively. Current Figures 4B and S4A indicate that Nup155 and FTSJ1 protein and mRNA show a repression release of approximately ~2 fold in the p53 siRNA conditions under Nutlin-3a treatment.

6) Figure S3, please indicate the number in each categories.
 Were only 5 candidates identified and assayed in this "mini-screen" ??
 Provide the list of proteins involved in translation as defined here by the authors.

The number of downregulated proteins upon Nup155 knockdown was n = 84 and out of these 8 proteins could in our opinion be linked to translation based on UNIPROT and GeneCards annotations. Besides the 5 candidates

we have included in our siRNA approach one could in principle also connect *Mitochondrial ribonuclease P protein 3* (uniprot ID O15091), *Thiosulfate sulfurtransferase* (uniprot ID Q16762), and *39S ribosomal protein L50, mitochondrial* (uniprot ID Q8N5N7) with translation. However, these candidates operate in the mitochondria and therefore we do not expect a primary involvement in cytoplasmic p21 mRNA translation. Based on the limited number of candidates we also decided not to use the term "siRNA screen" but "siRNA approach" in the revised manuscript. We are fully aware of the limitations of this small scale approach and assume that many more proteins are possibly involved in mRNA translation (for instance in the subgroup of upregulated proteins upon Nup155 knockdown or proteins with rather indirect impact on mRNA translation in both subgroups). Therefore, it was not our intention to claim to have included/identified all relevant proteins involved in the observed phenotype besides FTSJ1.

REVIEWERS' COMMENTS:

Reviewer #1 (Remarks to the Author):

The authors have done a reasonable job in revising the manuscript based on the suggestions that I originally made when I reviewed this study. I don't have any other comments.

Reviewer #2 (Remarks to the Author):

The authors have done impressive efforts to improve their manuscript in comparison to the original version. They have fully and satisfactorily addressed my concerns and suggestions. I have no further remarks.

Reviewer #3 (Remarks to the Author):

The authors have extensively addressed all the points raised in my previous review.

I have no other request, possibly one suggestion that would be to include a scheme with the feedback loop model, to make this very nice study more attractive to readers out of the field.

REVIEWERS' COMMENTS:

Reviewer #1 (Remarks to the Author):

The authors have done a reasonable job in revising the manuscript based on the suggestions that I originally made when I reviewed this study. I don't have any other comments.

Reviewer #2 (Remarks to the Author):

The authors have done impressive efforts to improve their manuscript in comparison to the original version. They have fully and satisfactorily addressed my concerns and suggestions. I have no further remarks.

Reviewer #3 (Remarks to the Author):

The authors have extensively addressed all the points raised in my previous review. I have no other request, possibly one suggestion that would be to include a scheme with the feedback loop model, to make this very nice study more attractive to readers out of the field.

Again we thank all reviewers for their supportive comments and are very happy that they appreciated our efforts to improve the manuscript during the revision process. As suggested by Reviewer #3 we have also included an illustration with the proposed feedback loop model (Figure 7D).